# EMBODIED WEB AGENTS: Bridging Physical-Digital Realms for Integrated Agent Intelligence

**Yining Hong*    Rui Sun*    Bingxuan Li†    Xingcheng Yao†    Maxine Wu†    Alexander Chien†**
**Da Yin    Ying Nian Wu    Zhecan James Wang    Kai-Wei Chang**

University of California, Los Angeles

## Abstract

AI agents today are mostly siloed — they either retrieve and reason over vast amount of digital information and knowledge obtained online; or interact with the physical world through embodied perception, planning and action — but rarely both. This separation limits their ability to solve tasks that require integrated physical and digital intelligence, such as cooking from online recipes, navigating with dynamic map data, or interpreting real-world landmarks using web knowledge. We introduce EMBODIED WEB AGENTS, a novel paradigm for AI agents that fluidly bridge embodiment and web-scale reasoning. To operationalize this concept, we first develop the EMBODIED WEB AGENTS task environments, a unified simulation platform that tightly integrates realistic 3D indoor and outdoor environments with functional web interfaces. Building upon this platform, we construct and release the EMBODIED WEB AGENTS Benchmark, which encompasses a diverse suite of tasks including cooking, navigation, shopping, tourism, and geolocation — all requiring coordinated reasoning across physical and digital realms for systematic assessment of cross-domain intelligence. Experimental results reveal significant performance gaps between state-of-the-art AI systems and human capabilities, establishing both challenges and opportunities at the intersection of embodied cognition and web-scale knowledge access. All datasets, codes and websites are publicly available at our project page https://embodied-web-agent.github.io/.

## 1    Introduction

Recently, we have seen the proliferation of web agents capable of retrieving information online [Shi et al., 2017, Yao et al., 2022, Deng et al., 2023, Zhou et al., 2023, Koh et al., 2024] — yet they remain confined to screens disembodied from the real world. Meanwhile, their physical counterparts — robots and embodied systems — navigate the world but with limited access to the Internet. What if the boundary between the digital and physical realms were shattered? What if web agents stepped out of the browser, with keys to perceive and act in the real 3D physical world, while physical robots autonomously tapped into the encyclopedic knowledge of the web? As illustrated in Figure 1, such agents would not only assess the ingredients in your kitchen, search for matching recipes online, shop for missing items, and cook your favorite dish for you; but also traverse historical landmarks, interpret architectural styles using both their own perception and Wikipedia, leave personalized reviews, and perhaps even return with a souvenir in hand. We, as humans, don't compartmentalize our intelligence into "physical-only" and "digital-only" modules — we fluidly move between realms. What if contemporary AI agents could likewise achieve the best of both worlds?

Building such agents *goes far beyond a mere combination of isolated web and embodied systems*; it presents a set of deeply intertwined challenges. The first is *the perceptual grounding problem*: how can an agent link abstract digital instructions (e.g., "cook potato and egg until golden brown" as in Figure 1 (b)) with the high-dimensional data streams of the physical world (e.g., visually recognizing

39th Conference on Neural Information Processing Systems (NeurIPS 2025) Track on Datasets and Benchmarks.

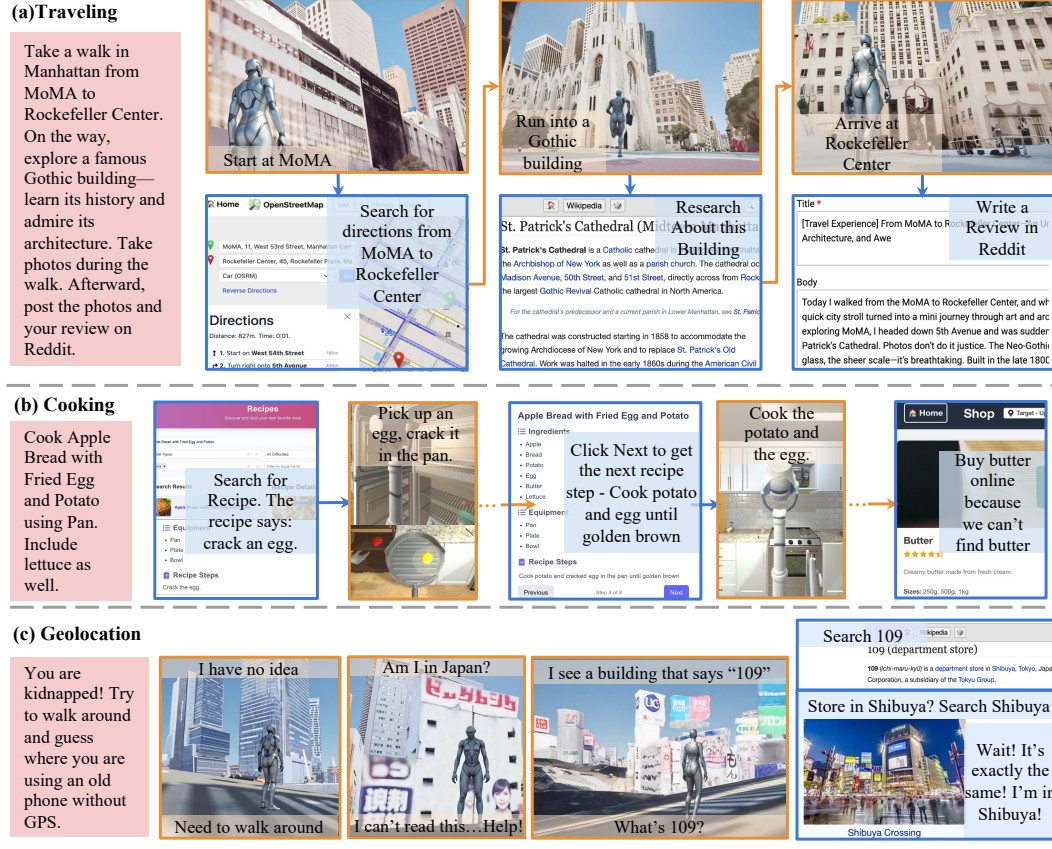

Figure 1: **Illustrative examples of our EMBODIED WEB AGENTS conceptual paradigm, tasks and environments.** Blue boxes and arrows indicate web interaction / switching to the web respectively. Orange boxes and arrows indicates acting in / switching to the embodied environment. We omit most intermediate actions due to the large number of interaction steps.

the transition of potatoes and eggs to a golden state through a series of embodied observations)? Addressing this requires embodied perception, where agents actively interpret their surroundings through movement, interaction, and multimodal sensing — continually acquiring feedback from their environment and aligning these observations with digital instructions. The second challenge is *cross-domain planning*: how should an agent decide when to shift between physical actions and digital information retrieval, particularly when information from one domain contradicts or supplements the other? For instance, the online map may suggest a path to visit Rockefeller Center, but real-world observation may reveal that the center is closed due to a protest, demanding a dynamic reevaluation of the agent's plan. To navigate seamlessly between domains, agents must maintain a coherent and persistent representation that bridges physical and digital contexts — recalling physical experiences when operating online, and retrieving digital knowledge when acting in the world. Despite all these challenges, there remains a surprising lack of research targeting this level of integrated intelligence — both in terms of conceptual frameworks and benchmark development. As a result, progress in each domain often unfolds in isolation, with limited cross-pollination between the two paradigms.

To this end, we introduce EMBODIED WEB AGENTS as a new conceptual paradigm of AI systems that unify physical embodiment with web-scale knowledge access — capable of perceiving and acting in the real world while reasoning over dynamic, unstructured information from the web. To operationalize this concept, we first develop the EMBODIED WEB AGENTS task environments, a unified simulation platform that integrates realistic 3D environments with interactive web interfaces. This platform combines (1) indoor settings from AI2-THOR, (2) outdoor navigation in Google Earth, and (3) web interfaces including Wikipedia, online stores, recipe websites, map services *etc.*, enabling agents to interact seamlessly with both physical and digital spaces. Building upon this environment, we construct the EMBODIED WEB AGENTS Benchmark, which encompasses approximately 1.5k

tasks across multiple domains, including: (1) cooking tasks where agents match physical ingredients with online recipes; (2) navigation combining online maps with physical wayfinding; (3) shopping requiring coordination between in-store actions and online options; (4) tourism connecting physical landmarks with web information; and (5) geolocation determining position through embodied exploration and online research. Together, these tasks systematically test an agent's ability to bridge embodied perception, action, and web-based reasoning across varied contexts.

We conduct comprehensive experiments on our proposed EMBODIED WEB AGENTS benchmark using several state-of-the-art LLM agent baselines, including GPT, Gemini, Qwen, and Intern models. Experimental results show that current LLM agents are far from satisfactory compared to human performances. A detailed breakdown and analysis of error types and their percentage contributions to task failures also reveal that current models predominantly struggle with cross-domain integration, not isolated capabilities. For instance, these models encounter problems such as being trapped in a single environment and unable to switch to the other domain, or the misalignment of web instructions and embodied actions. This further strengthens our position that embodied web agency presents unique challenges that cannot be studied through isolated physical or digital agents alone, as the key difficulties emerge precisely at the intersection where these domains are intertwined.

The key contributions of this paper can be summarized as follows.

- We introduce EMBODIED WEB AGENTS as a new conceptual paradigm for AI systems that integrate embodiment with web-scale information access — formalizing a class of agents capable of acting in the physical world while reasoning over unstructured digital content.
- We develop the EMBODIED WEB AGENTS task environments, a unified simulation platform that tightly integrates realistic 3D environments with interactive web interfaces, enabling agents to perform cross-domain tasks involving perception, action, and retrieval.
- We construct and release the EMBODIED WEB AGENTS Benchmark, which encompasses a diverse suite of tasks across multiple domains including navigation, shopping, traveling, cooking and geolocation.
- We conduct in-depth empirical analysis of state-of-the-art LLM agents on our benchmark, revealing that our benchmark poses rigorous challenges for current LLM agents, and opens up a challenging new direction and testbed for future agents with integrated intelligence.

## 2 Related Works

**Web Agent Benchmarks** Web agents are designed to navigate and interact with web environments to complete tasks following user instruction. Initial web agent evaluation benchmarks such as MiniWoB [Shi et al., 2017] and MiniWoB++ [Liu et al., 2018] introduce a suite of diverse web navigation tasks on synthetic webpages. More recent benchmarks emphasize greater realism and task diversity. WebShop [Yao et al., 2022] simulates an e-commerce platform with numbers of products to evaluate agents' ability to search and make purchases, while Mind2Web [Deng et al., 2023] provides a diverse collection of open-ended tasks across hundreds of real websites to assess general web navigation and interaction capabilities. Similarly, benchmarks like WebArena [Zhou et al., 2023], WebVoyager [He et al., 2024], WebLINX [Lù et al., 2024], and VisualWebArena [Koh et al., 2024] feature fully functional websites spanning multiple domains, enabling the evaluation of agents on long-horizon tasks in realistic, diverse environments. OVEN [Hu et al., 2023] challenges models to link images to specific Wikipedia entities given text queries. Beyond pursuing more realistic test environments, WorkArena [Drouin et al., 2024] requires agents to interact with enterprise software and perform tasks demanding higher expertise and comprehension. In this work, we explore a distinct yet important scenario where web browsing is integrated into the physical embodied world.

**Embodied Environments and Benchmarks** Recent developments in environments and benchmarks have accelerated the research on embodied AI. Simulation platforms, such as AI2-THOR [Kolve et al., 2017], Habitat [Manolis Savva* et al., 2019] and iGibson [Shen et al., 2021, Li et al., 2022], enable agents to perform diverse interactive tasks in realistic indoor environments. Benchmarks like ALFRED [Shridhar et al., 2020] and BEHAVIOR [Srivastava et al., 2021] provide a diverse suite of indoor tasks for embodied agents, requiring instruction understanding, long-horizon planning and manipulation in a closed environment. Additionally, Embodied Agent Inferface [Li et al., 2024] formalizes decision processes for LLM-based embodied agents and introduces fine-grained evaluation metrics for indoor embodied tasks. Efforts have also been made to extend the applicability

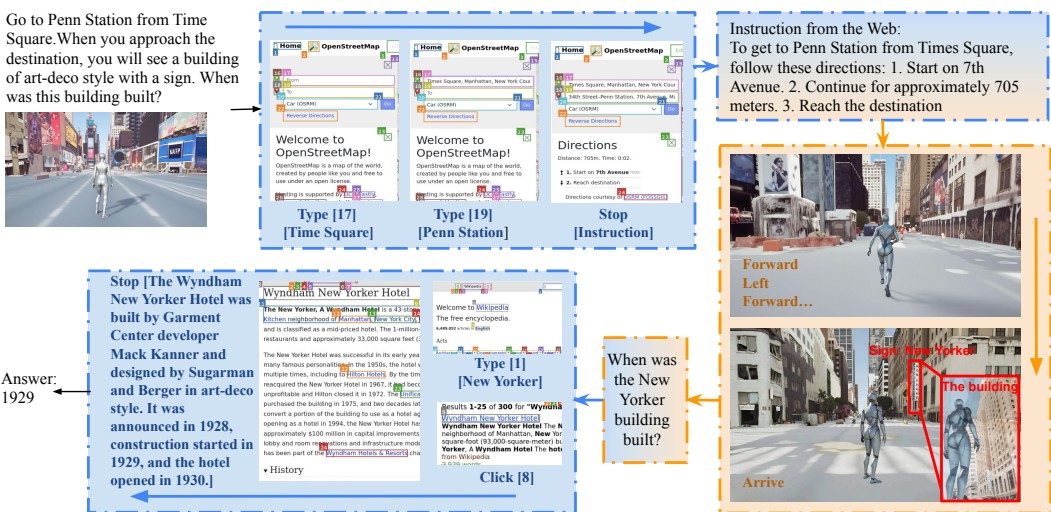

Figure 2: **An Exemplar Pipeline of completing a task in our EMBODIED WEB AGENTS dataset.** Blue boxes indicate web interaction. Orange boxes indicate embodied interaction. Boxes with gradient colors indicate switching from one environment to the other.

of embodied agents to outdoor environments. A series of outdoor navigation benchmarks, such as StreetLearn [Mirowski et al., 2018], TouchDown [Chen et al., 2019, Mehta et al., 2020], RUN [Paz-Argaman and Tsarfaty, 2019], have been introduced to evaluate the ability of embodied agents on vision-language navigation and spatial description resolution in urban street environments. Du and Varshney [2016], Du et al. [2019] create immersive systems that integrate geo-tagged social media with 3D street-level environments to enhance virtual and augmented reality experiences for storytelling, tourism, and cultural exploration. Yang et al. [2024] V-IRL is a platform for training and testing AI agents in realistic virtual environments to develop real-world skills. More outdoor related tasks such as geolocation prediction [Haas et al., 2023] and map understanding [Xing et al., 2025] has also been proposed recently. In this work, we design a new benchmark encompassing a diverse set of embodied tasks within both indoor and outdoor environments. Different from previous works, our benchmark focuses on embodied tasks that require web access and interaction to be completed, a realistic scenario that is challenging and neglected in existing benchmarks.

**Cross-Modal Agent Systems** Cross-modal agent systems integrating vision, language and other modalities have been explored in both web and embodied environments. In web-based settings, He et al. [2024] builds a web agent powered by a large multimodal model that interacts with real-world websites following user instructions. Lin et al. [2024] develops ShowUI, an efficient vision-laguage-action model for GUI agent. For embodied tasks, multimodal foundation models such as Gato [Reed et al., 2022], PaLM-E [Driess et al., 2023] and 3D-LLM [Hong et al., 2023] have been developed to provide generalist policies in real world. In this work, we explore a new dimension for modal fusion in embodied agents, by integrating both embodied and web actions into one unified framework, to enable agents to perform more complex and diverse tasks with real-world applications.

## 3 The EMBODIED WEB AGENT Task Environments

Inspired by Zhou et al. [2023], our environments are formalized as $E = \langle S, A, O, T \rangle$, where $S$ is the combined physical-digital state space, $A$ is the action space spanning both domains, and $O$ is the observation space comprising embodied input $o_t^e$ and web perception $o_t^w$. The deterministic transition function $T : S \times A \to S$ governs state evolution as agents select actions based on task specification, observations, and history. Task completion is measured by reward function $r(a_1^T, s_1^T)$ evaluating whether actions successfully fulfill intents like cooking dishes or reaching destinations.

Our task environments can be categorized into three parts: outdoor environment (3.1), indoor environment (3.2) and web environment (3.3). We show an example of interacting with and switching among the environments in Figure 2, as well as the action spaces of all environments in Table 1.

| Action | Explanation |
|--------|-------------|
| **INDOOR ENVIRONMENT ACTIONS** | |
| *Agent Movement* | |
| `Teleport [obj]` | Teleport agent to a specific object |
| `MoveAhead/Back/Left/Right` | Move agent in a cardinal direction |
| *Object Interaction* | |
| `PickupObject / PutObject [obj]` | Pick up or put held object |
| *Object State Changes* | |
| `OpenObject / CloseObject [obj]` | Open or close an object |
| `SliceObject [obj]` | Slice an object |
| `CookObject [obj]` | Cook an object |
| *Environment Switching* | |
| `switch_environment [msg]` | Switch between web/embodied |
| **OUTDOOR ENVIRONMENT ACTIONS** | |
| `Forward / Left / Right` | Move agent in outdoor environment |
| **WEB ENVIRONMENT ACTIONS** | |
| *Page Operation Actions* | |
| `click [id]` | Click on an element with specific id |
| `type [id] [content] [pr]` | Type content into field |
| `scroll [direction]` | Scroll page up or down |
| `hover [id] / press [key_comb]` | Hover or simulate key press |
| *Tab Management & URL Navigation Actions* | |
| `new_tab / close_tab / tab_focus` | Open, close or focus on a tab |
| `goto [url] / go_back / forward` | Navigate to URL or go back/forward |

Table 1: Action Spaces for All Environments

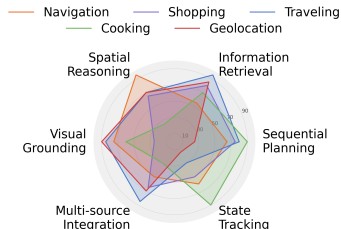

Figure 3: Importance of Different Capabilities Across Tasks

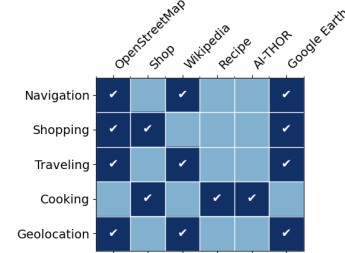

Figure 4: Environments for Tasks

## 3.1 Outdoor Environment

The outdoor environment is constructed by leveraging the Google Street View and Google Earth API, which provides real-world, street-level observations captured by Google's panoramic cameras. To build the outdoor environment, we select four cities (i.e., New York, Boston, Philadelphia, and Pittsburgh) with visually and structurally complex street layouts. Unlike synthetic or simulation-based environments, the visual data provided by Google is inherently more natural, noisy, and diverse, offering a more challenging and representative benchmark. Through API calls, we retrieve observations associated with specific geographic coordinates. These include panoramic images or standard-perspective images in cardinal directions. Alongside visual data, we also obtain: the GPS coordinates of each point, the heading / directional metadata between connected points, and the connectivity (adjacency) information across locations. With these elements, we construct a navigation graph that underlies the outdoor environment. Formally, this environment can be described as an undirected graph $G = (V, E)$, where each node $v \in V$ represents a specific GPS coordinate, each edge $e \in E$ encodes a connection between two coordinates, including heading and distance, and each node is associated with four directional visual observations (north, east, south, west), represented as standard field-of-view images. Agents interact with the outdoor environment by observing these visual inputs, accessing the neighboring node set, and using heading information to reason about spatial transitions. Given navigation instructions (e.g., derived from web-based directions), the agent must determine which neighbor to move to at each step in order to reach a specified goal location, completing the navigation task through step-by-step decision making. This design closely mirrors real-world settings and introduces challenges that go beyond those posed by synthetic simulators. Compared to environments with simplified or rendered visuals, our outdoor environment demands stronger generalization and robustness from embodied agents, making it a more practical and realistic testbed for evaluating agent systems in open-world scenarios.

## 3.2 Indoor Environment

The indoor task environment utilizes AI2-THOR [Kolve et al., 2017], a photorealistic 3D indoor simulation platform. The environment provides highly accurate and interactive kitchen scenes containing fresh ingredients, cooking equipment, storage containers, and kitchen appliances. Agents can observe ingredient states, manipulate objects, and monitor cooking progress through visual perception. Objects are tracked with properties and states, including boolean flags (e.g., isSliced, isCooked), location information (e.g., parentReceptacles), and more, all of which dynamically update as agents execute physical actions like chopping or mixing, instructed by online recipes. A specialized state evaluator compares the current kitchen state against ideal target states, measuring task completion by checking whether objects have achieved desired states and spatial arrangements.

### 3.3 Web Environment

The web environment consists of five functional websites, each supporting different aspects of agent interaction across both indoor and outdoor scenarios. The websites are implemented with a React.js frontend structured using modular components and state management, and a FastAPI backend that exposes asynchronous RESTful APIs for data serving and user interaction. The homepage serves as the central navigation hub, linking to all other task-specific websites and maintaining contextual continuity across interactions. The recipe website we built allows users to browse, search, and filter cooking recipes based on ingredients, dietary preferences, or cuisine types. The shopping website built from scratch enables management of a shopping cart, ingredient lookup, and simulated checkout processes. It facilitates task flows involving item selection, inventory reasoning, and purchasing. We also adapt several websites from the WebArena benchmark [Zhou et al., 2023]. The OpenStreetMap site offers an interactive map for location search, address lookup, and exploration of geographic entities. The Wikipedia site presents richly interlinked encyclopedic content for information-seeking, entity linking, and multi-hop reasoning across documents. These websites are modified slightly to ensure smooth integration with the homepage. All websites are public and can be reached. More details can be found in our project page `https://embodied-web-agent.github.io/`. We also include more details and screenshots of the web environment in the Supplementary Material.

## 4 The EMBODIED WEB AGENTS Benchmark Construction

In this section, we describe how we construct our EMBODIED WEB AGENTS benchmark. We will cover 5 domains of tasks: Navigation, Shopping, Traveling, Cooking and Geolocation. We show examples of the tasks in Figure 1, and a full pipeline of completing a task in Figure 2. Figure 3 summarizes the required level of each capability for successful task completion across domains, and Figure 4 shows which environments are utilized in different tasks.

**Navigation** Building upon the Outdoor Environment described in § 3.1, our navigation tasks evaluate an agent's spatial reasoning ability to reach destinations based on web-sourced directions. We use the OpenStreetMap website in § 3.3 to ensure reproducibility and consistent web interaction. To create diverse navigation scenarios, we prompt GPT-4o-mini to generate geographic coordinates across the aforementioned cities. These coordinates serve as either the start or end points of a task, and the graph structure centered around each point can be developed using our outdoor environment. During the prompting process, we also generate initial task instructions tied to the obtained coordinates. After identifying start or end points, we locate the corresponding counterparts using node adjacency relationships in the outdoor graph, forming a path within the environment. For evaluation purposes, we compute the shortest path using Dijkstra's algorithm as our ground-truth trajectory.

Navigation tasks require bidirectional interaction between web and embodied domains. The agent must input origin and destination into the map website to obtain directions, then ground these instructions in the embodied environment through turning actions and movements. Our benchmark includes 144 navigation tasks, each requiring both web interaction and embodied navigation. Since VLM-generated locations may have connectivity issues or misalignments with actual map data, we conduct human verification for all tasks to ensure their correctness and validity.

**Shopping** In real life, when buying products, we typically compare prices online, decide where to purchase based on pricing and store location information, place an order online, and then visit a physical store for pickup. Our shopping tasks evaluate the agent's ability to handle both online shopping and embodied environment interactions. The agent must place orders through our self-hosted shopping website dicussed in § 3.3, obtain store locations, and navigate in the outdoor environment to the correct store for pickup using the directions by OpenStreetMap; alternatively, it may also first navigate to a store and then place the order online.

In our benchmark, we simulate four stores located in distinct areas of Manhattan, New York. Our website lists a variety of items with product names, images, prices, and store information including distance and store name. The agent needs to *weigh both the price of the item and the store's location to make an optimal decision*, ultimately grounding web information into the embodied environment and navigating to the store for the selected item. To generate diverse scenarios, we design multiple templates with different items and user intents, which are listed in detail in our Supplementary Material. We also test the agent's ability to retrieve information across multiple browser tabs—*e.g.*, requiring the agent to complete a purchase, return to the homepage, switch to a map website, and

search for directions before embodied navigation. Some complex tasks require multiple rounds of web interaction and physical navigation within a single shopping scenario, testing agents' multi-source integration and sequential planning abilities. In total, our dataset contains 216 shopping tasks.

**Traveling** Inspired by how people consult web resources while traveling to navigate the physical world more effectively, we include traveling as a primary benchmark task. Using our custom-built outdoor environment and a pipeline similar to navigation tasks, we prompt a VLM to generate starting points, destinations, and initial task instructions, which we then refine into detailed, context-appropriate versions. Unlike pure navigation tasks that focus on following map directions and resolving map-reality inconsistencies, traveling tasks emphasize richer interaction between web resources and the embodied environment. For instance, when an agent encounters a significant landmark during navigation, as shown in Figure 1 (a) when it runs into a Gothic building, it may query Wikipedia to retrieve relevant information about that location. The agent is also expected to explore different architectural styles or historical landmarks, and ground Wikipedia descriptions to physical observations (*e.g.,* grounding the text descriptions of appearances of a Gothic building to the actual observation of the building). Web interactions in traveling tasks extend beyond map reading to include diverse informative sites, creating scenarios with multiple intertwined interactions between digital and physical domains. Our benchmark includes 110 traveling tasks, each requiring fluid movement between embodied navigation and web-based information retrieval.

**Cooking** As described in § 3.2, we use AI2-THOR as our indoor environment. To generate embodied cooking tasks for execution, we begin by identifying all ingredients available in the AI2-THOR kitchen scenes. We then manually search online for recipes that include these ingredients. Since online recipes are often noisy and may not align with the constraints of the AI2-THOR environment, we use Claude to refine them. Claude is guided by a predefined set of allowable agent actions in AI2-THOR environment to ensure the resulting recipes are executable. To increase task difficulty, we introduce confounders for most of the recipes by including pairs of recipes with the same name but differing in difficulty level, dietary type, ingredients used, or required cooking equipment. The users can filter out recipes based on these constraints by filter bars below the search bar (as in our self-hosted websites discussed in § 3.3). The next step is to curate a set of tasks based on collected recipes. For each scene, we retrieve recipes that match the available ingredients. The task instruction asks the agent to cook the corresponding dish. When a confounder exists for a given recipe, we introduce additional constraints — *e.g.*, "Diet type is vegetarian," "Use a tomato," — to disambiguate between recipe variants. If an ingredient does not exist in the scene, the agent is expected to go online to shop for it. The cooking tasks evaluate the agent's capability to perform long-trajectory planning in the indoor environment, and continuously check if the states match with the web instruction in the process. Our benchmark contains in total 911 cooking tasks. An exemplar task is in Figure 1 (b).

**Geolocation** Geolocation is a classic computer vision task Hays and Efros [2008], where models predict geographic coordinates of given images. Instead of treating it purely as a conventional vision problem, we reinterpret it based on its inherent characteristics as an embodied geolocation task. Inspired by the design of GeoGuessr, we move away from the single-image input setting and treat the model as an agent situated in an embodied environment. The agent is allowed to explore the outdoor environment we construct and ultimately output its estimated location. During exploration, the agent interprets storefront texts, visual cues, and street-view observations while accessing web information when needed to supplement its observations. The agent explores these environments freely, performing web interactions when additional information is needed. The task concludes when the agent has either 1) explored all possible positions or 2) collected sufficient information to confidently predict its location. This framework unifies embodied navigation, web-based reasoning, and visual grounding into a cohesive geolocation task. Our data collection is adapted from Huang et al. [2025], focusing on examples from existing geolocation datasets where models typically fail. We select coordinates where we hypothesize web information may improve prediction accuracy, then construct environments centered on these points using Google API. Geolocation evaluates the visual grounding ability of agents. An example is shown in Figure 1 (c). We collect 142 such data.

## 5   Experiments

In this section, we first introduce baseline LLM agents (§ 5.1) and evaluation metrics (§ 5.2) we use for experiments. We then perform result analysis (§ 5.3) on our EMBODIED WEB AGENTS benchmark. We group the results of Navigation, Shopping and Traveling together as they are all

related to outdoor planning. Please refer to the Supplementary Material for more experimental results, experimental setup, LLM prompts, qualitative examples and error cases, as well as more analyses.

## 5.1 Baseline LLM Agents

We evaluate four LLMs as our baseline agents: GPT-4o-mini, Gemini 2.0 Flash, Qwen-VL-Plus, and InternVL-2.5-latest. GPT-4o-mini is OpenAI's state-of-the-art multimodal model with strong performance in visual reasoning and real-time interaction. Gemini 2.0 Flash, by Google DeepMind, is optimized for speed and efficiency while maintaining robust vision-language capabilities. Qwen-VL-Plus, from Alibaba's Qwen Team, offers fine-grained image-text understanding. InternVL-2.5-latest, developed by Shanghai AI Lab, excels in spatial and semantic reasoning.

## 5.2 Evaluation Metrics

To comprehensively assess agent performance across physical and digital domains, we employ four evaluation metrics for outdoor planning and cooking: **Overall Accuracy** measures the success of complete cross-domain task execution, requiring both successful web task completion (reaching the terminal web state) and fulfillment in the embodied environment, representing holistic task completion that necessitates seamless integration of both domains; **Web-only Accuracy** evaluates the ability to successfully complete the web portion of a task, such as reaching the final step of a recipe, isolating digital domain independent of physical execution; **Embodied-only Accuracy** assesses an agent's ability to achieve all required physical state conditions in the embodied environment, such as properly slicing ingredients, or navigating to a desired place, measuring physical domain proficiency; and **Overall Completion Rate** represents the proportion of task progress achieved, indicating how much of the required state conditions have been fulfilled relative to the total task objectives.

## 5.3 Result Analysis

| | | Task / Metric | GPT | Gemini | Qwen | Intern | Human |
|---|---|---|---|---|---|---|---|
| **Outdoor Tasks** | **Navigation** | Overall Accuracy | 34.72 | 30.56 | 15.97 | 13.19 | 90.28 |
| | | Overall Completion Rate | 52.08 | 48.96 | 36.81 | 26.04 | 91.32 |
| | | Web-only Accuracy | 69.44 | 67.36 | 57.64 | 38.89 | 92.36 |
| | | Embodied-only Accuracy | 48.61 | 46.53 | 31.25 | 23.61 | 90.97 |
| | **Shopping** | Overall Accuracy | 25.46 | 23.61 | 13.89 | 10.65 | 92.59 |
| | | Overall Completion Rate | 31.94 | 30.56 | 18.52 | 14.35 | 93.52 |
| | | Web-only Accuracy | 39.35 | 37.50 | 23.15 | 17.13 | 93.06 |
| | | Embodied-only Accuracy | 34.26 | 32.41 | 17.59 | 12.96 | 93.98 |
| | **Traveling** | Overall Accuracy | 30.91 | 25.45 | 11.82 | 9.09 | 91.82 |
| | | Overall Completion Rate | 50.91 | 48.18 | 34.55 | 20.91 | 93.64 |
| | | Web-only Accuracy | 57.27 | 53.64 | 41.82 | 25.45 | 94.55 |
| | | Embodied-only Accuracy | 47.27 | 44.55 | 29.09 | 19.09 | 92.73 |

Table 2: **Model Performance Across Different Outdoor Tasks.** There is a huge performance gap between LLM agents' performances and human performances.

| Metric | Vision | | | | Text | | | | Human |
|---|---|---|---|---|---|---|---|---|---|
| | GPT | Gemini | Qwen | Intern | GPT | Gemini | Qwen | Intern | |
| Overall Acc | 5.4 | 4.1 | 0.6 | 0.0 | 6.4 | 5.8 | 1.5 | 0.4 | 77.08 |
| Completion Rate | 40.26 | 35.62 | 15.91 | 9.73 | 39.16 | 38.92 | 17.20 | 10.02 | 85.37 |
| Web Acc | 59.71 | 47.74 | 28.65 | 10.64 | 57.08 | 62.23 | 35.89 | 15.58 | 100 |
| Embodied Acc | 8.7 | 6.1 | 2.2 | 0.9 | 10.5 | 8.2 | 4.1 | 1.3 | 77.08 |

Table 3: **Model Performance for Cooking Task.** The models achieve inferior overall accuracies.

**Outdoor Planning** For outdoor planning, we use GPT-4o-mini alongside Gemini 2.0 Flash, Qwen-VL-Plus, and InternVL-2.5-latest to evaluate performance across navigation, shopping, and traveling tasks (Table 2). For web observation, we follow the setting of VisualWebArena. We observe that: 1) GPT-4o-mini consistently leads across all metrics, with the highest accuracy in navigation (34.72%), shopping (25.46%), and traveling (30.91%), though still well below human performance. Gemini follows closely behind, while Qwen and Intern lag behind. 2) Web-only accuracy exceeds

embodied-only accuracy for all outdoor tasks, suggesting models handle digital information more effectively than physical navigation. 3) Generally, completion rates are satisfactory, while overall accuracies are very low across all tasks. This indicates models can execute parts of complex tasks but struggle with consistent cross-domain reasoning over longer sequences. 4) From task perspective, shopping and traveling involve richer interactions between the embodied environment and the web than navigation, and each task spans longer steps. As a result, the overall accuracy for shopping and traveling is noticeably lower than for navigation. This highlights the difficulty of cross-environment tasks, particularly those that are lengthy and involve multiple steps, for current models.

**Cooking** For cooking, we implement two distinct approaches: vision-based and text-based. Our vision-based implementation draws inspiration from VisualWebArena, utilizing screenshot images of websites enhanced with Set-of-Marks (SoM) annotations that highlight interactive elements. For embodied observations, we provide first-person visual perspectives from the agent's viewpoint within the AI2-THOR environment. The text-based implementation follows WebArena's methodology, representing web content through accessibility trees that capture the semantic structure of websites in textual form. For embodied observations, we extract structured scene graphs directly from AI2-THOR, providing explicit object relationships and states. We use Qwen-PLUS and InternLM-latest for Qwen and Intern models without vision.

Table 3 presents performance metrics for various models on the cooking task, comparing vision-based and text-based approaches against human performance. A substantial performance gap exists between AI models and humans, with the best model (text-based GPT-4o-mini) achieving only 6.4% overall accuracy compared to humans' 77.08%. Text-based models using structured scene graphs consistently outperform their vision-based counterparts using first-person views, suggesting current models struggle to ground visual observations effectively in cooking contexts. GPT-4o-mini and Gemini-2.0-Flash demonstrate substantially stronger performance than Qwen-VL-Plus/Qwen-PLUS and InternVL/InternLM across both modalities. Notably, similar to outdoor performances, all models perform significantly better on web-only tasks compared to embodied-only tasks, revealing that while current models can navigate recipe websites effectively, they struggle with physical execution requiring object manipulation and state tracking. Despite low overall accuracy, models achieve moderate completion rates, indicating partial task success but failure in full cross-domain integration.

**Geolocation** For geolocation tasks, we benchmark against FairLocator [Huang et al., 2025], a study analyzing VLM performance on GeoGuessr using Google Street View images. As shown in Table 4, the embodied web agent, capable of active exploration and web information access, significantly outperforms the passive baseline, particularly in identifying finer-grained locations like cities and streets. We observe consistent improvements across all models when moving from the baseline to embodied setting, suggesting the performance gains are model-agnostic. Interestingly, we also find that even when the retrieved Wikipedia search results are noisy or uninformative, the act of querying itself often helps the agent reason more confidently. This indicates that formulating search queries may serve as a form of self-supervision. This substantial improvement underscores the potential of integrating embodied and web domains to enhance performance across numerous real-world tasks, warranting further investigation.

| | Setting / Model | Continent | Country | City | Street | All |
|---|---|---|---|---|---|---|
| **Geolocation** | **FairLocator** | | | | | |
| | GPT-4o-mini | 90.85 | 81.69 | 73.24 | 1.41 | 1.41 |
| | Gemini-2.0-Flash | 93.66 | 85.92 | 78.17 | 0.70 | 0.70 |
| | Qwen-VL-Plus | 76.06 | 58.45 | 45.07 | 0.70 | 0.00 |
| | InternVL2.5-Latest | 77.46 | 62.68 | 52.11 | 1.41 | 1.41 |
| | **Embodied Web Agent** | | | | | |
| | GPT-4o-mini | 97.18 | 90.85 | 85.21 | 3.52 | 3.52 |
| | Gemini-2.0-Flash | 97.18 | 94.37 | 85.21 | 4.23 | 4.23 |
| | Qwen-VL-Plus | 80.28 | 69.01 | 49.30 | 0.00 | 0.00 |
| | InternVL2.5-Latest | 93.62 | 77.30 | 57.45 | 2.13 | 1.42 |

Table 4: **Model performance for geolocation task.** All models performed much better when predicting after interactively exploring the environment and querying the web than just using static images.

## 5.4 Error Analysis

Figure 5 presents a detailed breakdown of error types and their percentages that contribute to task failures in cooking tasks when using GPT-4o. Our analysis reveals that the primary challenges in embodied web agents lie not in isolated capabilities, but in their integration. While embodied errors (14.6%) and web errors (8.0%) occur, cross-domain errors (66.6%) overwhelmingly dominate the failure landscape — confirming that the critical bottleneck emerges at the intersection where physical and digital domains meet. The most prevalent failure pattern involves agents becoming trapped in single-domain cycles. In 23.6% of failures, agents get stuck in the embodied environment, repeatedly executing irrelevant physical actions without

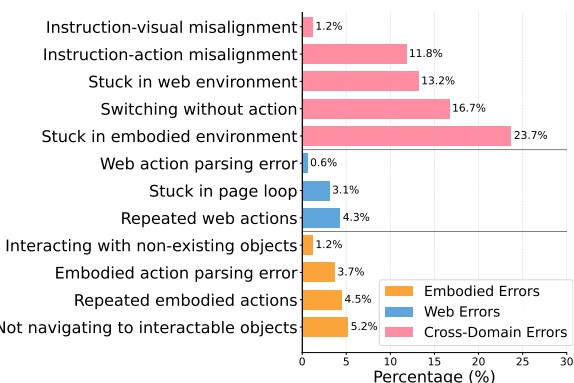

Figure 5: **Error Analysis for Cooking Tasks.** We can see that the majority of errors are cross-domain errors.

returning to the web for the next step. Similarly, in 13.2% of cases, agents remain fixed in web environments, endlessly clicking "next" through recipe pages without initiating cooking actions. In addition, agents often switch between environments without meaningful action (16.7%) or suffer from instruction-action misalignments (11.8%), such as slicing lettuce when a recipe instructs "slice the apple". Web interaction failures manifest as agents getting stuck in page loops (3.1%) or performing identical actions repeatedly (4.3%). In the embodied domain, agents fail to navigate to interactable objects (5.2%) or execute repeated actions (4.5%). These isolated domain errors are far less frequent than cross-domain integration failures, explaining why LLM agents achieve only 6.4% overall accuracy despite moderate performance on single-domain tasks. This confirms that embodied web agency presents unique challenges requiring focused research on mechanisms that bridge physical and digital reasoning.

## 6 Conclusion

In this paper, we introduced EMBODIED WEB AGENTS, a new paradigm for AI research that bridges the artificial divide between physical and digital intelligence. Through our comprehensive benchmark spanning cooking, navigation, shopping, tourism, and geolocation tasks, we demonstrate that current AI systems face significant challenges in fluidly integrating embodied perception with web-based information retrieval. These findings establish a foundation for future research in integrated intelligence systems, highlighting the need for developing AI agents that can seamlessly traverse physical and digital worlds. A limitation is our reliance on simulated agents, which may not fully capture the complexity and unpredictability of physical-digital interactions of real robots.

## Broader Impact

Our EMBODIED WEB AGENTS research presents both opportunities and challenges for society. On the positive side, agents that bridge physical and digital domains could enhance accessibility for individuals with mobility limitations, support contextualized learning environments, and improve emergency response through integrated information access. However, several risks warrant attention. First, these agents may exhibit "dual-domain hallucination," where errors propagate across physical and digital realms, compounding misinformation. Second, systems that connect physical environments with web platforms introduce novel privacy concerns beyond those in either domain alone.

To mitigate these concerns, our benchmark provides transparent evaluation protocols that can identify cross-domain errors. We designed our environments as simulations that don't interact with real-world systems, limiting immediate risks while providing valuable research insights. By releasing our benchmark to the research community, we aim to encourage the development of more robust embodied web agents with improved error detection mechanisms before deployment in real-world settings.

## Acknowledgment

We thank anonymous reviewers for their helpful comments. This work was partially supported by U.S. DARPA ECOLE Program No. #HR00112390060, ONR grant N00014-23-1-2780, DARPA ANSR program FA8750- 23-2-0004, Amazon, Google and Apple Research Award. Chang was supported in part by a grant from DARPA to the Simons Institute for the Theory of Computing.

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
