# Supplementary Material for
# EMBODIED WEB AGENTS: Bridging Physical-Digital Realms for Integrated Agent Intelligence

**Yining Hong\*** **Rui Sun\*** **Bingxuan Li†** **Xingcheng Yao†** **Maxine Wu†** **Alexander Chien†**
**Da Yin** **Ying Nian Wu** **Zhecan James Wang** **Kai-Wei Chang**

University of California, Los Angeles

## Contents

39th Conference on Neural Information Processing Systems (NeurIPS 2025).

# A  Contribution Statement

Yining Hong is responsible for coming up with the idea; overall organization of the team (meetings; pointing out research directions; divison of responsibilities; reaching out to potential collaborators etc.); all the data, codes and experiments of indoor cooking; the majority of paper writing; creating demo videos.

Rui Sun implemented interactions between the outdoor environment and the web, made corrections to the environment, collected data for outdoor tasks (i.e., navigation, traveling, and shopping), and validated the data collection process. Rui also wrote part of the paper about outdoor tasks. He also wrote detailed instructions for implementing the Geolocation tasks for Maxine and Alexander.

Bingxuan Li took care of designing and implementing all the web environments used in this paper. Bingxuan also made the showcase website of this paper, and wrote the web development part of the paper.

Xingcheng Yao built the basic outdoor environment using the Google Street View API, which lays a good foundation for further development. Xingcheng also wrote part 2 and part 3 of related works.

Maxine Wu and Alexander Chien were responsible for the entire Geolocation section. Maxine implemented the baseline pipeline, contributed to the design of evaluation metrics, performed error analysis, and created the demo videos. Alexander implemented and contributed to the design of the embodied environment and the web interaction system for the final pipeline, curated the dataset, and wrote the corresponding sections of the paper. Maxine also adapted the pipeline to support different models. Both authors performed experiments in both the baseline and the embodied settings.

Da Yin came up with first-step instructions of the Geolocation task. He also wrote the web agent part of the related works.

Yingnian Wu, Zhecan James Wang and Kai-Wei Chang took the advising roles. Specifically, Prof. Wu provided initial insights on agent planning. Zhecan James Wang helped sort out the meeting notes and discussion results into documents; provided ideas on task design; helped coordination among people. Prof. Chang scheduled biweekly meetings with the team, gave valuable advice and pointed out valuable research directions, as well as helped polish the paper.

# B  Broader Impacts

Our EMBODIED WEB AGENTS research presents both opportunities and challenges for society. On the positive side, agents that bridge physical and digital domains could enhance accessibility for individuals with mobility limitations, support contextualized learning environments, and improve emergency response through integrated information access. However, several risks warrant attention. First, these agents may exhibit "dual-domain hallucination," where errors propagate across physical and digital realms, compounding misinformation. Second, systems that connect physical environments with web platforms introduce novel privacy concerns beyond those in either domain alone.

To mitigate these concerns, our benchmark provides transparent evaluation protocols that can identify cross-domain errors. We designed our environments as simulations that don't interact with real-world systems, limiting immediate risks while providing valuable research insights. By releasing our benchmark to the research community, we aim to encourage the development of more robust embodied web agents with improved error detection mechanisms before deployment in real-world settings.

# C  Dataset Statistics

In Figure 1, we show the detailed distribution of all tasks. In Figure 2, we show more statistics of the indoor cooking task, including the number of ingredients the task takes, the number of recipe steps as well as the distribution of diet types and difficulty levels.

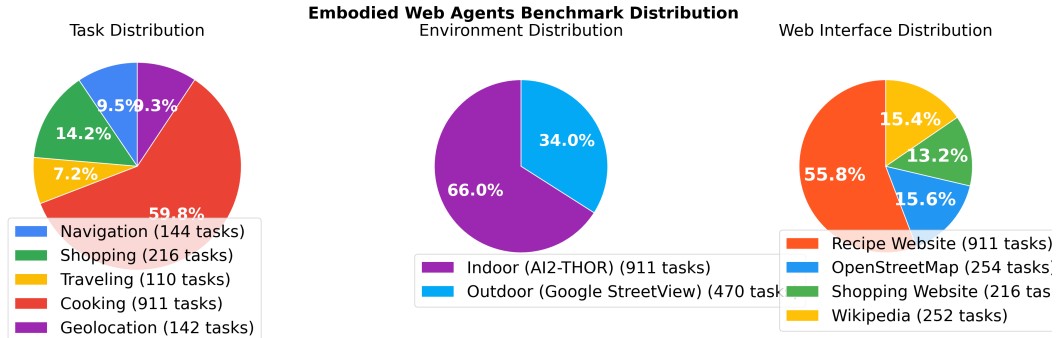

Figure 1: Task, scene and web distributions of our data

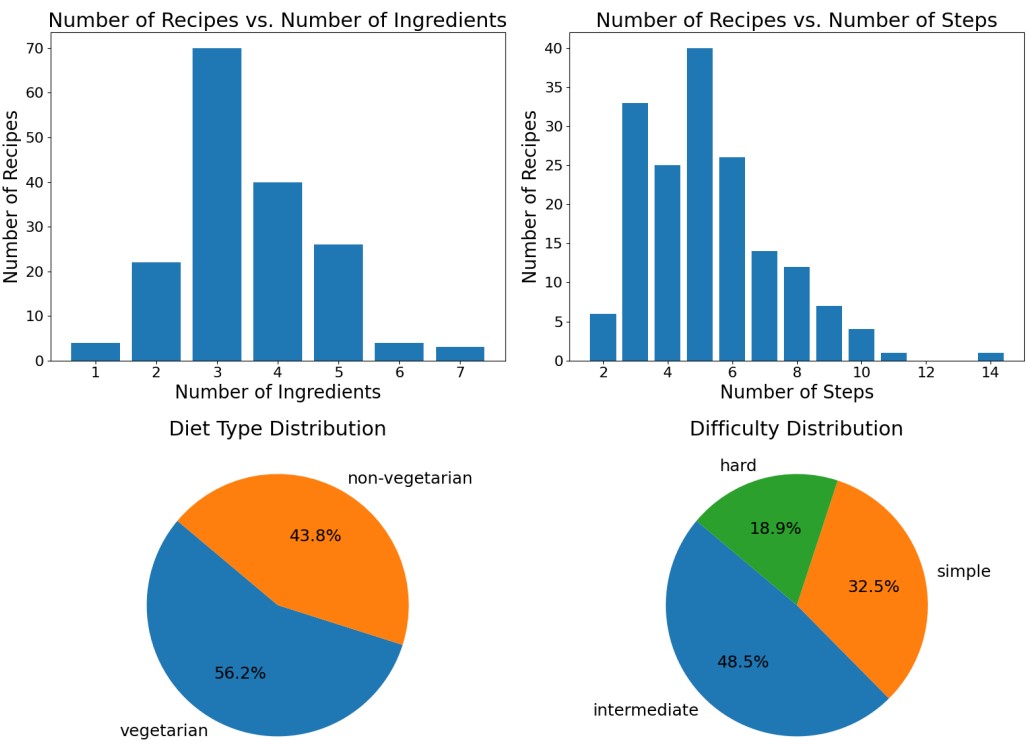

Figure 2: More data statistics of the indoor setting

# D More Details about Data Collection

## D.1 Outdoor Data Collection

**Task intents and templates.** To better organize and summarize task intents, we've designed a set of task templates. Each template corresponds to several specific tasks. Below is an overview of these task templates.

> **Templates for Outdoor Tasks**
>
> **Navigation:**
> * Show me the fastest route from {origin} to {destination}.
> * Plot a walking path from {origin} to {destination}, avoiding highways.
> * Starting at {origin}, guide me step-by-step to {destination}.
> * If I leave {origin} right now, what's the quickest way to reach {destination}?
> * ...
>
> **Shopping:**
> * Add all the items with {{quality}} on this page into my cart.
> * Add something like the {{item}} to my shopping cart.
> * Buy me a {{product}} with {{detail}}.
> * Can you add {{item}} {{condition}} to my wishlist?
> * How many calories are in {{item}} and {{secondary_item}}? I need to select a lower one.
> * What size of {{item}} should I buy if {{condition}}?
> * ...
>
> **Wikipedia:**
> * Search for "{query}" and tell me more about it.
> * I would like to know more about "{query}".
> * Look up "{query}".
> * Provide me more information about "{query}".
> * ...

In these templates, the placeholders in {} will be replaced with actual content. For example, "Show me the fastest route from {origin} to {destination}" might become "Show me the fastest route from the Penn Station to Times Square".

**Task and location generation.** Although we have a list of templates for task intents, we still need to prompt the VLM to generate the initial task intent. Along with the task intent, since our dataset is a combination of embodied and web task, we also need to generate the location from our outdoor environment then we can proceed to the next step. The prompt to generate task and location can be seen in Section F.1.

**Annotation Tool.** To better support data annotation and visualization, we designed and built our own annotation tool shown as Figure 3. Having this graphical interface makes manual inspection and correction much simpler. Moreover, because our tasks involve outdoor navigation, we frequently need to visualize trajectories on a map, which lets us view the results in a very intuitive way. We can also directly update the data by using the annotation tool. When we modify the data, we save the changes made in the annotation tool's interface directly to the backend JSON file.

## D.2 Geolocation Data Collection

**Dataset Curation.** The dataset we present is composed of samples from the Breadth dataset of FairLocator. We randomly sample 142 locations and manually review each to ensure they are reasonable—meaning they contain enough visual cues for a model to make a prediction.

**Image Observations.** To collect image observations, we use the Google Street View API to obtain views at our ground-truth coordinates. To support exploration in our embodied pipeline, we also query nearby "adjacent" viewpoints—defined as nodes within one edge in the Street View panorama graph, typically corresponding to a small translation from the initial location. For this reason, during sampling, we also exclude locations without adjacent viewpoints.

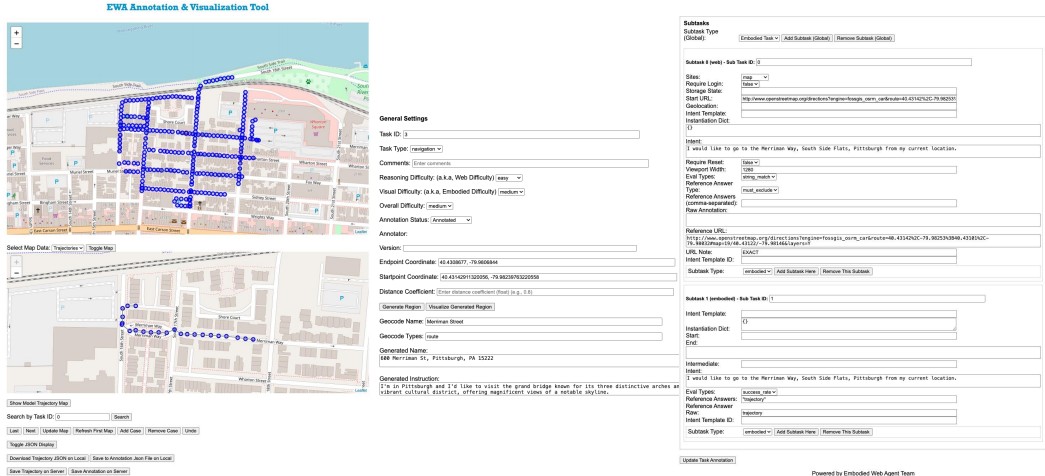

Figure 3: Annotation tool. Here is the annotation tool interface. It features three map windows side by side: one visualizes the coordinate points within the outdoor environment, the second shows the ground-truth outdoor navigation path, and the third displays the agent's actual trajectory after navigation. Beyond simply visualizing points, you can directly edit the annotation JSON right in this interface. We load the JSON's contents into the front end; once human verification is complete, any needed edits can be applied by clicking "Update Task Annotation", which pushes your changes back into the backend JSON file. This gives you both a clear visual overview and a streamlined labeling-and-verification workflow.

**Website Queries.** We utilize VisualWebArena to do the website queries. Since the VisualWebArena environment requires configuration files specific to each query or intent, we dynamically generate a new configuration file each time the agent creates a new search prompt. We do not use predetermined configuration files, as we want to evaluate the agent's ability to use visual cues and identify its own knowledge gaps. Thus, the queries and configurations for each run are random and unique to the model in some sense. We also stipulate that queries should be styled simplistically and be optimized for Wikipedia searches since we only allow the model to access the Wikipedia site within our web environment. We concatenate these queries and their results in a context cache to feed to the agent during confidence estimations and the final prediction.

# E Qualitative Examples

## E.1 Outdoor Planning

The outdoor planning consists of three core subtask types, that are navigation, traveling, and shopping. Here, we present four illustrative examples for navigation error, traveling success, shopping error, and shopping success.

In Figure 4, the agent misinterprets complex map directions and moves in the wrong direction within the outdoor environment, ultimately causing the navigation task to fail. This highlights the agent's current limitations and biases when reading, processing, and grounding intricate routing information in the real world.

In Figure 5, it is a representative traveling task that the agent correctly processes the directions, reaches the designated location. Then, since the location is a point of interest, the agent queries Wikipedia and retrieves the correct information. This case demonstrates how the agent seamlessly integrates information from both web sources and the embodied environment to complete the entire traveling workflow.

In Figure 6, the agent fails to understand the function of certain web elements and, based on its vision-language input, does not ground its decision to the correct action (clicking versus typing). This failure exposes areas for improvement in the agent's action grounding capabilities, especially for web interactions. It also reveals weaknesses in its visual grounding when the clickable target is not visually salient.

In Figure 7, the agent successfully navigates to the correct store, processes the online shopping interface, and ultimately selects and purchases the right product. This success case illustrates the agent's ability to coordinate web-derived information with real-world movement to complete the full shopping task.

Together, these examples vividly illustrate the challenges and progress in grounding web information within embodied tasks across navigation, traveling, and shopping scenarios.

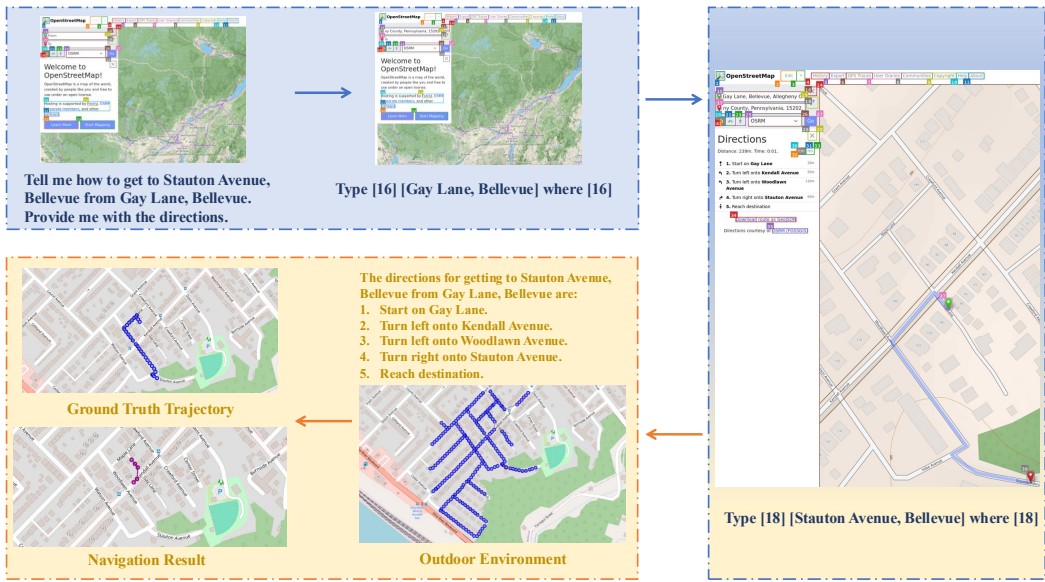

Figure 4: Navigation error. The agent's failure to correctly understand the directions from the map website led to navigation errors in the outdoor environment.

## E.2 Indoor Cooking

We show a full example of carrying out a cooking task following web instructions in Figure 12 and 13. As we can see, the model needs to perform multi-step iterative reasoning between the web side and embodied side to complete a complex cooking task. In Figure 14, we show a failure case. It fails because: 1) action grounding error. The web instruction is to slice apple and bread. However, it also tries to crack the egg. 2) Stuck in the embodied side and cannot go back to the web side. When it fails to crack an egg, it starts to perform random actions in the embodied environment without trying to go back to the web environment.

## E.3 Geolocation

Figure 8 illustrates a step-by-step example of the embodied geolocation pipeline. We begin with images from the initial standpoint—one image facing each of the four cardinal directions. The agent uses these observations to generate a web query, formulated in the style of a Wikipedia search. This query is executed using the VisualWebArena environment and the resulting web page content is retrieved. Both the image observations and the web search results are then passed to the agent's confidence estimation module, which assesses whether the current context is sufficient for making an accurate geolocation prediction. In this example, the agent initially determines that it lacks sufficient confidence. It then chooses to move to another nearby adjacent standpoint, gathers new image observations, and re-evaluates its confidence. Upon receiving the additional context, the agent is

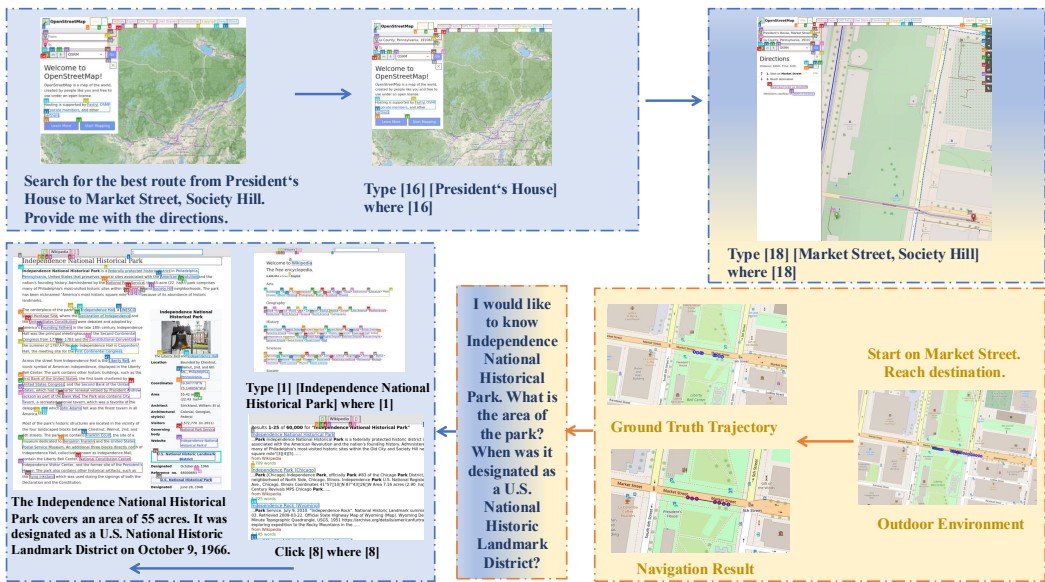

Figure 5: Traveling success. The agent first correctly understood the user's request and provided accurate map directions. It then navigated through the outdoor environment and moved to the correct location. Because this was a traveling task at a tourist site, the agent finally queried the environment by consulting the Wikipedia page, obtained the right information, and successfully completed the entire task.

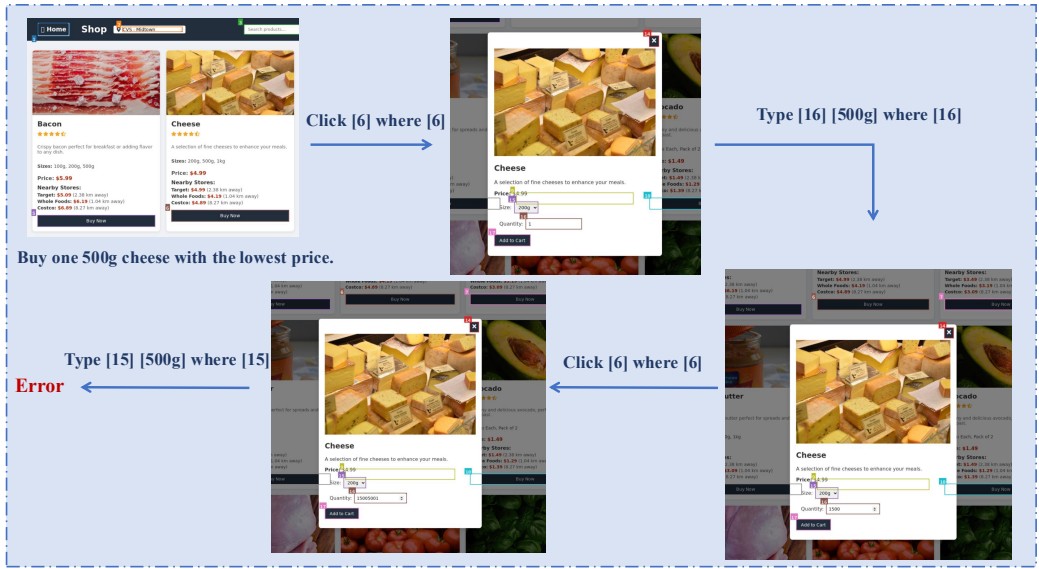

Figure 6: Shopping error. The agent failed to correctly interpret the elements on the webpage and likewise did not produce the correct action based on its visual and language inputs (it should have clicked instead of typing), which ultimately caused the agent to err and fail to complete the shopping task.

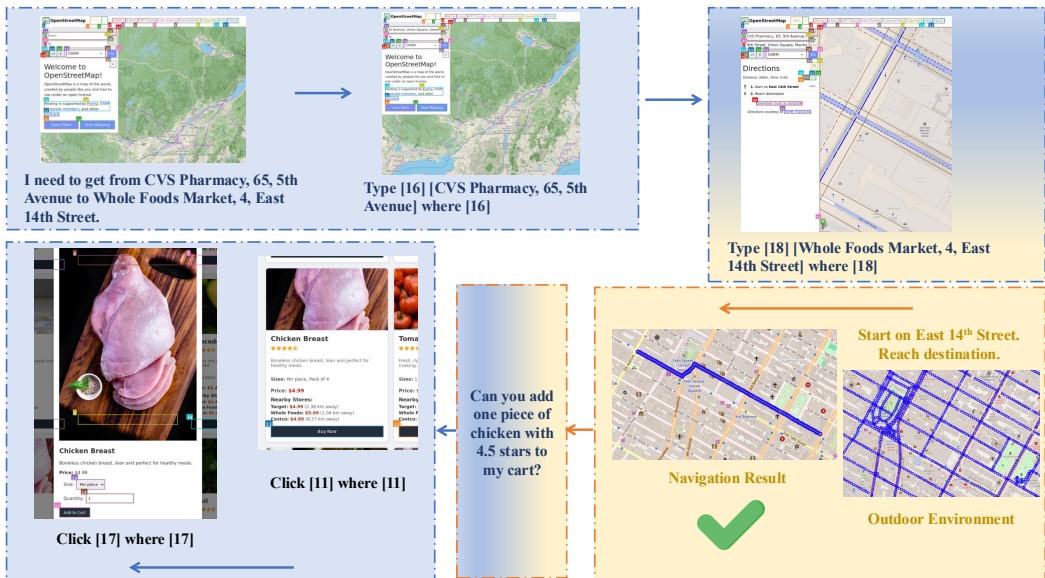

Figure 7: Shopping success. The agent successfully completed all of its subtasks. First, it retrieved the correct directions from the webpage, then navigated to the right location and began shopping. Finally, during the shopping process, it selected the correct item and saw the entire shopping task through to completion.

confident enough to make a prediction and proceeds to output both its predicted location and its reasoning.

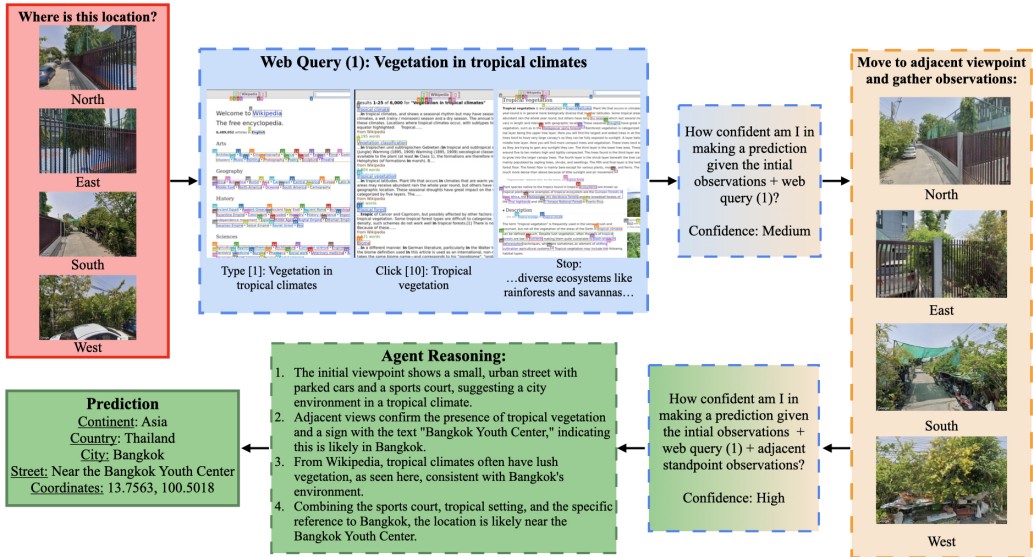

Figure 8: An exemplar pipeline of completing the geolocation task. The red box indicates initial input, the blue box indicates web interaction, the orange box indicates embodied interaction, and the green boxes indicate agent reasoning and prediction. Boxes with gradient colors indicate switching from one environment to the other.

# F LLM Prompts

## F.1 Outdoor

Here we list three key prompts used in our outdoor tasks. First, the prompt for generating locations at the very beginning. Second, based on those generated locations and the initial instruction, we use a task generation prompt to create the detailed subtasks, both embodied and web-based. After generating each task, we perform human verification and the verified data then becomes the dataset we use for our experiments. Once the experiments begin, the web-interaction portion still relies on the same prompts used in VisualWebArena. For the outdoor navigation portion, we employ the outdoor navigation prompt shown below. For visualization purposes, some of the prompts shown have been appropriately shortened.

---

**Location Generation for Outdoor Tasks**

You are an AI assistant that is familiar with cities all around the world. For a given city, please provide an iconic locations. For each location, provide a navigation instruction that captures its unique characteristics, historical significance, cultural importance, or architectural features WITHOUT directly mentioning its name. The navigation instructions should be specific enough that someone knowledgeable about the city could identify the exact location. Return the results as a JSON list where each element contains 'location' (a searchable address for geocoding) and 'instruction' (an informative instruction that avoids using the location's name but will uniquely locate the location). REMEMBER that the location MUST be in the city of New York, USA; Philadelphia, USA; Boston, USA; Pittsburgh, USA!

Here are four examples:
```
["location": "350 5th Ave, New York, NY 10118", "instruction": "I'm
in New York City and I'd like to go to the Art Deco skyscraper from
the 1930s that held the title of world's tallest building for nearly
40 years. It has 102 stories and is an enduring symbol of the city's
ambition."]
["location": "520 Chestnut St, Philadelphia, PA 19106",
"instruction": "I'm in Philadelphia and I'd like to go to the
red-brick Georgian hall with a white steeple in the historic district
where revolutionary delegates gathered in the 18th century to debate
and adopt the nation's founding documents."]
["location": "4 Jersey St, Boston, MA 02215", "instruction": "I'm
in Boston and I'd like to go to the century-old ballpark that
opened in 1912, famous for its emerald-green left-field wall and
as the longstanding home of one of Major League Baseball's oldest
franchises."]
["location": "601 Commonwealth Pl, Pittsburgh, PA 15222",
"instruction": "I'm in Pittsburgh and I'd like to go to the
150-foot-tall water jet fountain at the tip of downtown's triangular
park, marking where three rivers converge against a backdrop of the
city skyline."]"
```

Only give me one spot in one of these cities: New York, USA; Philadelphia, USA; Boston, USA; Pittsburgh, USA.

---

**Task Generation for Outdoor Tasks**

You are an Embodied Web Agent capable of obtaining information from webpages and executing tasks within an embodied environment. I will provide you with a generated_instruction and a generated_name from the embodied environment. The generated_instruction is a description of the task that outlines what the embodied agent needs to do, while the generated_name is the name or address of a location that the embodied agent needs to go to.

Based on the generated_instruction and generated_name, you need to generate tasks that both the web and the embodied agent must execute. These tasks should ensure that after execution, the embodied web agent can obtain information from the web to assist in completing the task in the embodied environment.

We have four types of webpages, and I will provide you with a task category. Based on the task category, you need to generate one or more tasks that interact with the webpages. The tasks must make use of at least one of the following webpages (or multiple): Task categories include: Shopping, Navigation, and Traveling.

The descriptions for these task categories are as follows:
1. Shopping: You need to search for product information, prices, store locations on the web. Then compare this information to find the most suitable store. Finally, the outdoor embodied agent can use the store address from the web to reach the store.
2. Navigation: You need to search for maps and route planning on the web. These details will help the outdoor embodied agent find the best route from the current location to the destination.
3. Traveling: You need to search for tourist attractions, travel guides, local culture, etc., on the web. Then, this information will help the outdoor embodied agent plan an itinerary and choose attractions or activities.

The types of webpages include: Shopping, OpenStreetMap, Wikipedia, and Homepage.
Here are descriptions of these webpages:
1. Shopping: This is a shopping website that provides information on various products, including prices and store locations. You can look for detailed product information and purchasing options here.
2. OpenStreetMap: This is an OpenStreetMap website, which provides maps and route planning services. You can search for your current location, destination, and best routes here.
3. Wikipedia: This is a Wikipedia website that provides encyclopedic knowledge on various topics. You can look up tourist attractions, local culture, travel guides, and more.
4. Homepage: This is a homepage website that provides links to the above websites. These websites can also lead you back to this homepage, making it convenient for users to switch between different websites.

Below are three examples. You need to generate output in this format:
Example 1:
Task Category: Traveling
generated_instruction: I'd like to visit the iconic Central Park, a sprawling urban park in New York City, known for its picturesque landscapes, recreational activities, and cultural landmarks. generated_name: Central Park, New York, NY
web_task_intent_0: Search for information about Central Park on the Wikipedia website. I would like to know about its open hours.
embodied_task_intent_1: I would like to explore Central Park and its various attractions.
web_task_intent_2: Find the best walking route between my current location and Central Park using the OpenStreetMap website.

Example 2:
Task Category: Navigation
...

Example 3:
Task Category: Shopping
...

You are given the following inputs:
Task Category: [task_category_placeholder]

generated_instruction: [generated_instruction_placeholder]
generated_name: [generated_name_placeholder]

Please only generate the embodied_task_intent and web_task_intent below.

---

### Outdoor Navigation

You are an embodied navigation agent operating within a street-view graph environment.

Each environment is defined by:
(1) A source node (starting latitude-longitude).
(2) A target node (destination latitude-longitude).
(3) A set of graph nodes, each with:

 (a) A unique node ID (lat-lng string).
 (b) Four street-view images (north, east, south, west) as your visual observations.
 (c) A list of neighbor nodes with absolute heading, descriptive text, and edge distance.

Your Objective:
Navigate step-by-step from the source to the target node by selecting exactly one neighbor at each step, according to the given parsed navigation instructions and the visual/textual context.

Available Inputs:
(1) Current node ID (string).
(2) Target node ID (string).
(3) Current absolute heading (degrees clockwise from true north).
(4) Parsed instructions: a list of action, distance pairs (action ∈ straight, left, right).
(5) Remaining distance (meters) to complete the current instruction step.
(6) List of previously visited node IDs (to avoid loops).
(7) For each neighbor:

 (a) Neighbor ID.
 (b) Absolute heading (°).
 (c) Relative heading to your current facing (°).
 (d) Distance (m) along the edge.

(8)Four visual observations: street-view images facing north, east, south, and west.

Your Task:
Based on all of the above, choose exactly one neighbor ID that best:
1. Follows the current action instruction (straight/left/right) relative to your facing.
2. Moves you toward the target by reducing distance.
3. Does not revisit an already visited node.

Response Format:
Reply with exactly one node ID (lat-lng string) on a single line, with no additional commentary.

---

## F.2 Geolocation

We design separate prompting strategies for the baseline and embodied pipelines. Additionally, we found that Qwen required significantly stricter prompt constraints to produce output consistent with our expected format, so we created dedicated prompts for Qwen.

In the baseline setting, we only prompt the vision-language model (VLM) with a single north-facing image from the initial standpoint. As shown in Figure 9, we use one prompt for GPT and Gemini and a separate version tailored to Qwen.

In contrast, the embodied pipeline involves multiple types of prompts, as illustrated in Figure 10. We use distinct prompts to:

- instruct the agent to move to adjacent standpoints,
- estimate confidence based on the current context,
- and generate a final location prediction.

The web query prompt (Figure 11) is issued after each new round of observations. The web query is executed in VisualWebArena and we add the results to a growing context cache along with prior image observations and web results. This evolving context is provided to the agent for both confidence estimation and the final prediction.

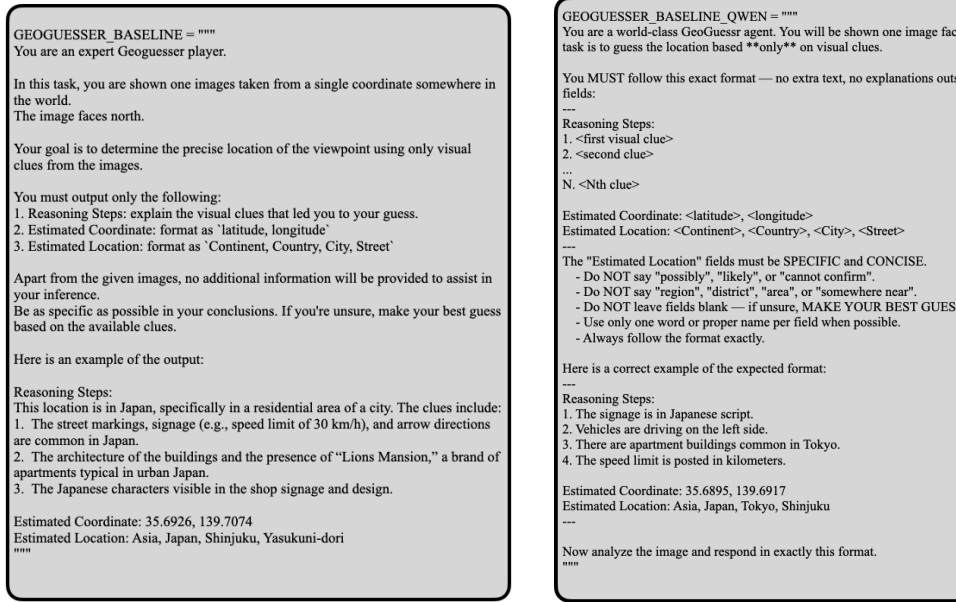

(a) GPT, Gemini baseline prompt.    (b) Qwen baseline prompt.

Figure 9: LLM prompts for geolocation baseline.

## G  Web Environment

In Figure 15, we show screenshots of our web environment. Please go to http://98.80.38.242:1220/ for more details.

## H  Human Performance

To establish meaningful benchmarks for our evaluations, we recruited undergraduate and graduate student volunteers from UCLA's Computer Science and Statistics departments. Participants were selected to represent a diverse range of technological familiarity and task-specific expertise. Each volunteer participated in a 2-hour session where they completed the same set of tasks that were presented to the AI models, covering both web-based and embodied scenarios in shopping, traveling, and cooking domains.

Our analysis reveals remarkable human performance across all domains, with overall accuracy rates ranging from 77.08% to 92.59%, significantly outperforming even the most capable AI systems. Particularly noteworthy is the consistent human performance across both web-based and embodied tasks, whereas AI models showed dramatic performance drops in embodied scenarios. This performance gap underscores the substantial challenges remaining in developing AI systems that can match human-level understanding and execution of everyday tasks that require multimodal reasoning, real-world knowledge application, and adaptive problem-solving strategies in response to environmental feedback.

GENERATE_ACTION = """
You are an expert GeoGuessr player.

Your task is to decide the next best move to help identify the exact location of the **initial viewpoint**.

Initial Viewpoint ID: {initial_id}
Current Viewpoint ID: {current_id}

You may choose from the following valid actions:
{actions}

- Use "Move[<ID>]" to explore a specific adjacent viewpoint.

---

Your output **must follow this format exactly**, with no additional commentary:

Reasoning Steps:
1. <reason 1>
2. <reason 2>
...

Action: <your chosen action from the list above>
"""

(a) Action generation prompt.

ESTIMATE_CONFIDENCE = """
You are an expert GeoGuessr player.

Your task is to estimate how confident you are in your ability to pinpoint the location of the **initial viewpoint**, based on **all available evidence**.

There are two main types of evidence:
1. **Visual observations** from multiple viewpoints (the initial and any visited adjacent locations). Each includes four images facing North, East, South, and West.
2. **Web search results** (if any), retrieved using Wikipedia-style queries that provide geographic or cultural context to support ambiguous clues.

---

Your output **must follow this exact format**, with no extra commentary:

Reasoning Steps:
1. <Visual clue or insight from the images>
2. <Clarification or confirmation from a web search result>
3. <Another clue from an adjacent viewpoint or supporting web info>
...

Confidence: <High | Medium | Low>

---

Confidence should reflect the following:
- **High** - You are very confident and could now make a highly accurate final guess.
- **Medium** - You've gathered some useful clues, but are still missing key evidence.
- **Low** - You lack clear regional indicators or need more support to make a good guess.

Now estimate your confidence using the clues presented below:
"""

(b) Confidence estimation prompt.

GENERATE_PREDICTION = """
You are an expert GeoGuessr player.

You have been shown a set of image observations and, optionally, some web search results from Wikipedia.
Each image observation comes from a specific location (the initial viewpoint or an adjacent one) and includes four directions: North, East, South, and West.

Your goal is to estimate the **precise location** of the initial standpoint, based on:
- Visual cues from all visited viewpoints (e.g. architecture, languages, terrain, signage)
- Insights derived from Wikipedia-style web searches (e.g. road sign conventions, license plates, vegetation)

---

Your output **must follow this format exactly**, with no additional commentary:

Reasoning Steps:
1. <Observation from image at initial viewpoint>
2. <Corroboration from adjacent views or directional variation>
3. <Clarifying information retrieved via Wikipedia (if used)>
4. <Final synthesis of all clues to narrow to a region or place>

Estimated Coordinate: <latitude>, <longitude>
Estimated Location: <Continent>, <Country>, <City>, <Street>

---

Be specific and logical in your reasoning.
Explain how your guess was informed by each clue and how web data helped disambiguate any uncertainty.
"""

(c) GPT, Gemini prediction prompt.

GENERATE_PREDICTION_QWEN = """
You are a world-class GeoGuessr agent. You have been shown a set of image observations and, optionally, some web search results from Wikipedia.
Each image observation comes from a specific location (the initial viewpoint or an adjacent one) and includes four directions: North, East, South, and West.

Your goal is to estimate the **precise location** of the initial standpoint, based on:
- Visual cues from all visited viewpoints (e.g. architecture, languages, terrain, signage)
- Insights derived from Wikipedia-style web searches (e.g. road sign conventions, license plates, vegetation)

You MUST follow this exact format — no extra text, no explanations outside the specified fields:

---
Reasoning Steps:
1. <first visual clue>
2. <second clue>
...
N. <Nth clue>

Estimated Coordinate: <latitude>, <longitude>
Estimated Location: <Continent>, <Country>, <City>, <Street>
---

The "Estimated Location" fields must be SPECIFIC and CONCISE.
   - Do NOT say "possibly", "likely", or "cannot confirm".
   - Do NOT say "region", "district", "area", or "somewhere near".
   - Do NOT leave fields blank — if unsure, MAKE YOUR BEST GUESS.
   - Use only one word or proper name per field when possible.
   - Always follow the format exactly.

Here is a correct example of the expected format:

---
Reasoning Steps:
1. The signage is in Japanese script.
2. Vehicles are driving on the left side.
3. There are apartment buildings common in Tokyo.
4. The speed limit is posted in kilometers.

Estimated Coordinate: 35.6895, 139.6917
Estimated Location: Asia, Japan, Tokyo, Shinjuku
---

Now analyze the image and respond in exactly this format.
"""

(d) Qwen prediction prompt.

Figure 10: LLM prompts for geolocation task exploration, confidence estimation, and final predictions.

```
GENERATE_WEB_QUERY = """
You are an expert GeoGuessr player.

Your goal is to generate a **Wikipedia-compatible web search query** — but only if it will help reduce your
uncertainty about the location you're seeing.

Observations:
- You've seen multiple viewpoints, each with four images (North, East, South, West).
- These images may contain road signs, languages, landscapes, architecture, vehicles, vegetation, and terrain.

Only ask about clues that you **don't fully understand** or recognize.

You are only allowed to search **Wikipedia** — so your question should be direct, factual, and focused on named
entities or structured categories.

Good Wikipedia-style questions:
- *"List of countries that drive on the left"*
- *"License plates in Southeast Asia"*
- *"Architecture of Scandinavian houses"*
- *"Languages using the Devanagari script"*
- *"Road sign color schemes by country"*
- *"Vegetation in Mediterranean climates"*

**Avoid vague or open-ended queries. Be specific.**

---

Here are queries you've already asked:
{query_history}

Rules:
+ Do **not** repeat a query that appears in the query history.
+ Focus on **new visual clues** and ask questions that Wikipedia can answer directly.
+ Use **titles, lists, or region-specific descriptions** common on Wikipedia pages.

---

Your output **must follow this format exactly**, with no additional commentary:

Reasoning Steps:
1. <visual clue you observed>
2. <why you need to look it up>

Intent Template: < general question form, e.g., "List of countries that use <element>"
Element: <the subject of the query, e.g., "Cyrillic script">
Intent: <the final question, e.g., "List of countries that use the Cyrillic script">
String Note: <what kind of answer you want, e.g., "A list of countries or regions that use the Cyrillic script in public
signage.">

---

Now generate a Wikipedia query using the observations below:
"""
```

Figure 11: LLM prompt for geolocation task web query generation.

**Task: Cook Ultimate Delight Sandwich with Egg. Diet Type: Non-Vegetarian. Difficulty: Hard**

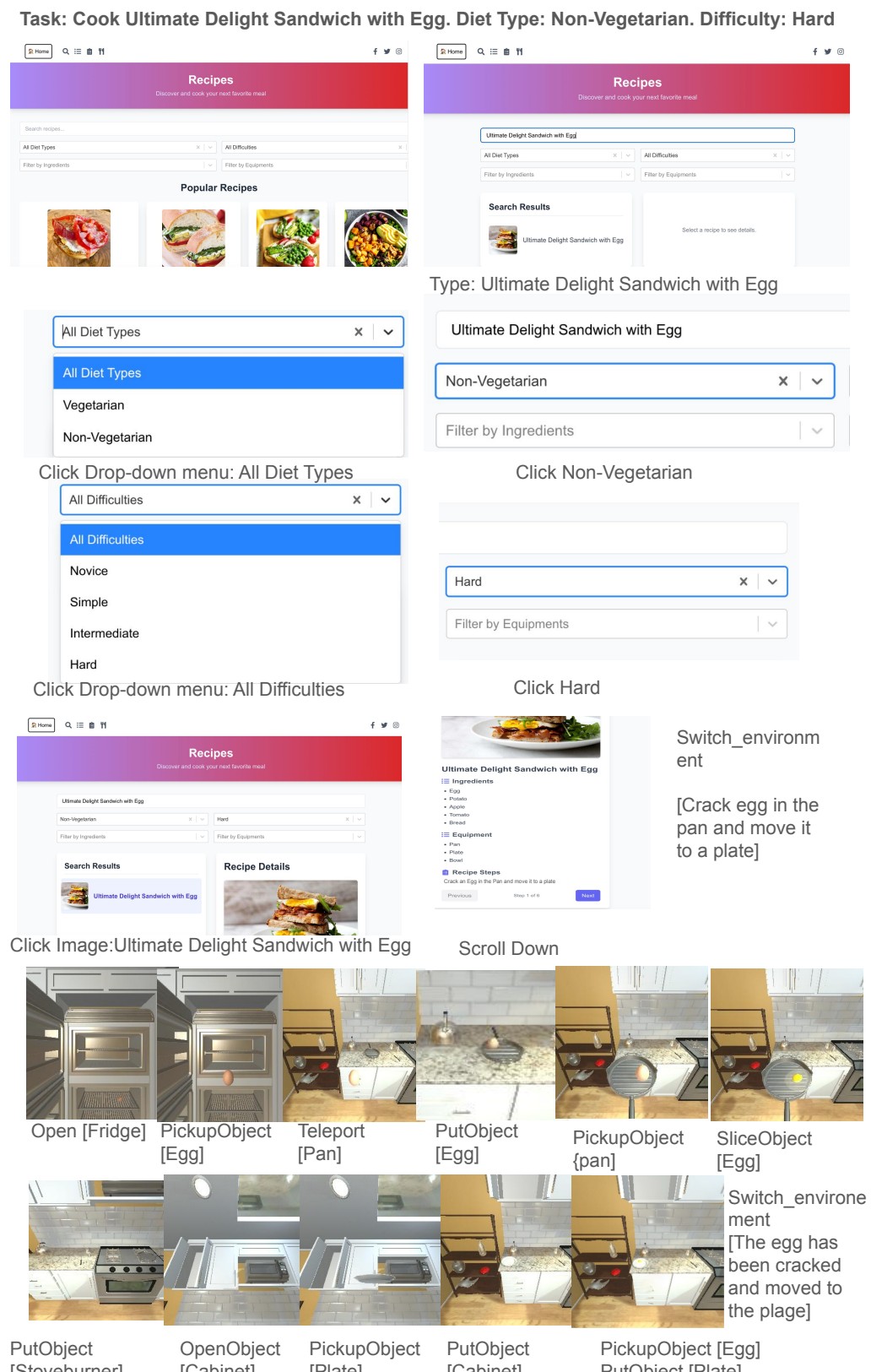

Figure 12: Qualitative example of indoor cooking - Part 1

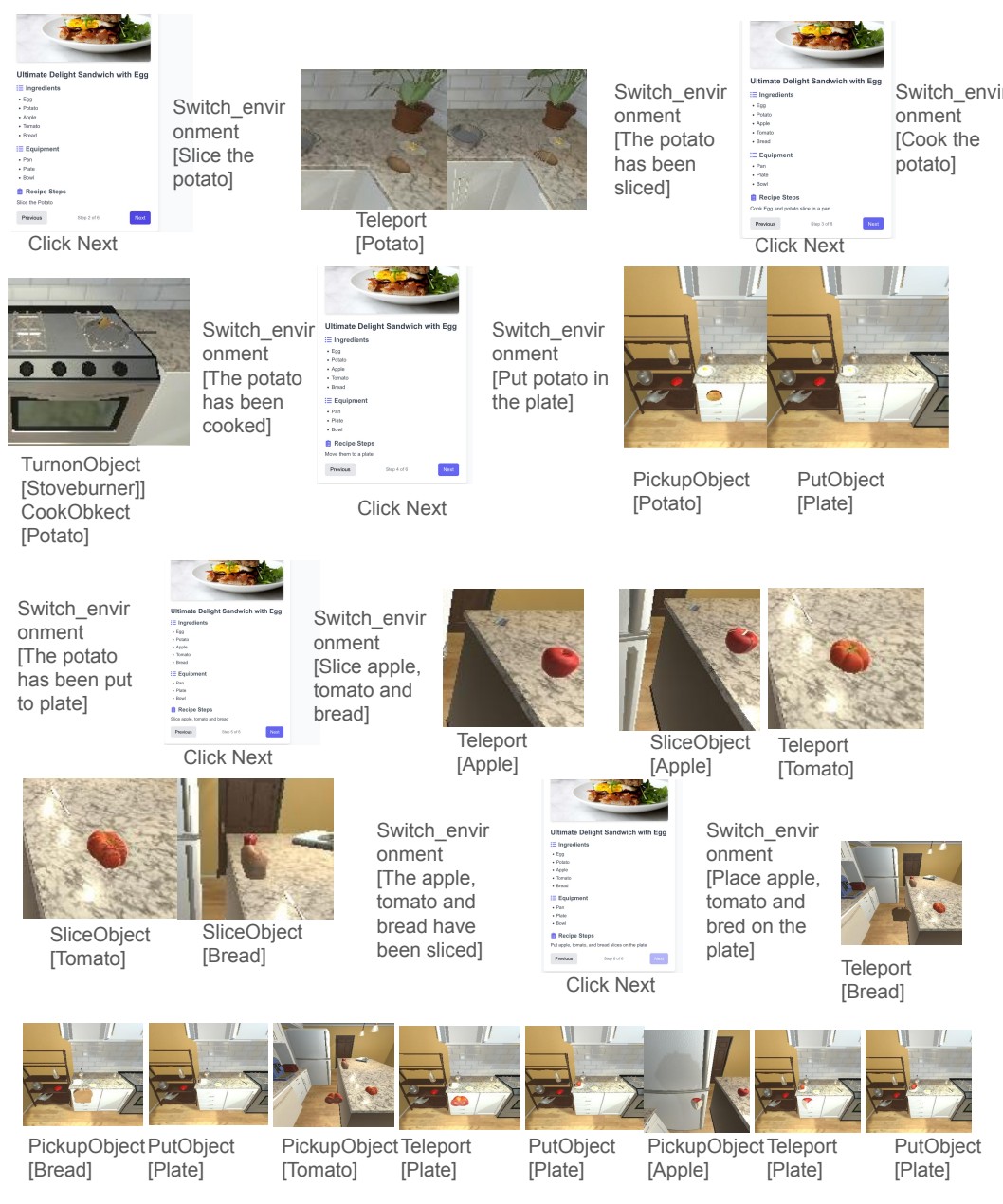

Figure 13: Qualitative example of indoor cooking - Part 2

**Task: Cook Scrambled Egg with Apple and Toast**

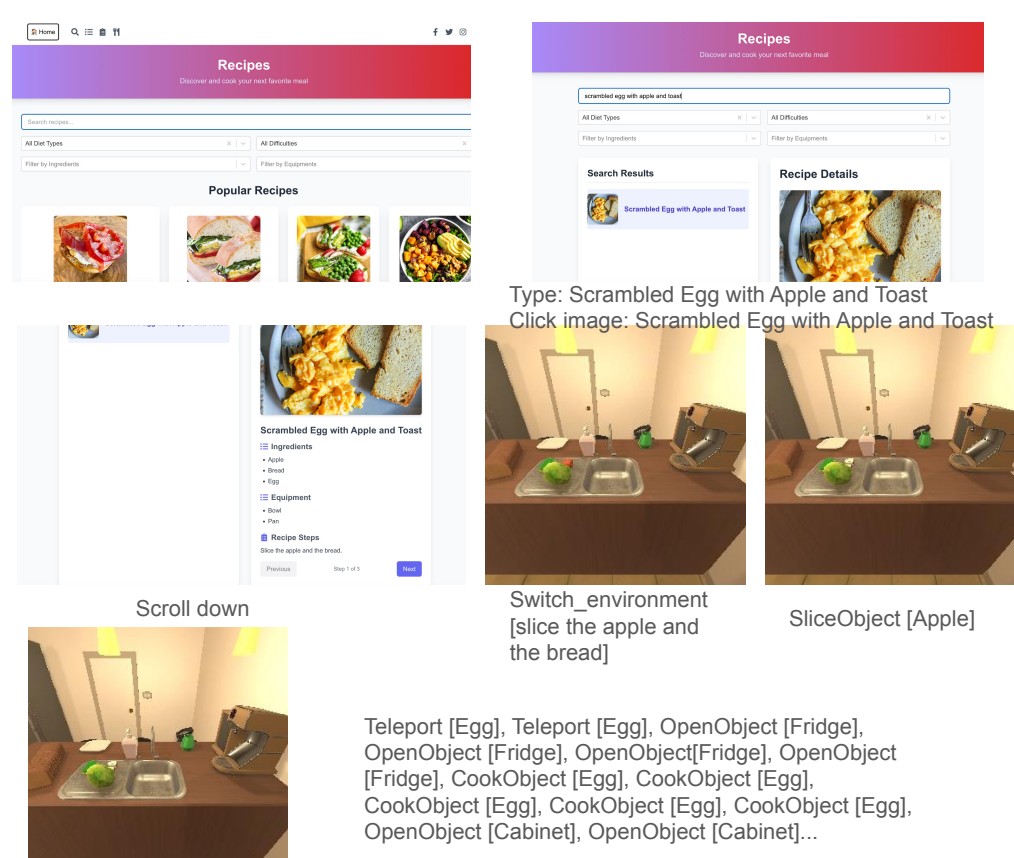

Type: Scrambled Egg with Apple and Toast
Click image: Scrambled Egg with Apple and Toast

Scroll down

Switch_environment
[slice the apple and
the bread]

SliceObject [Apple]

SliceObject [Bread]

Teleport [Egg], Teleport [Egg], OpenObject [Fridge],
OpenObject [Fridge], OpenObject[Fridge], OpenObject
[Fridge], CookObject [Egg], CookObject [Egg],
CookObject [Egg], CookObject [Egg], CookObject [Egg],
OpenObject [Cabinet], OpenObject [Cabinet]...

Failed

Figure 14: Failure case of indoor cooking

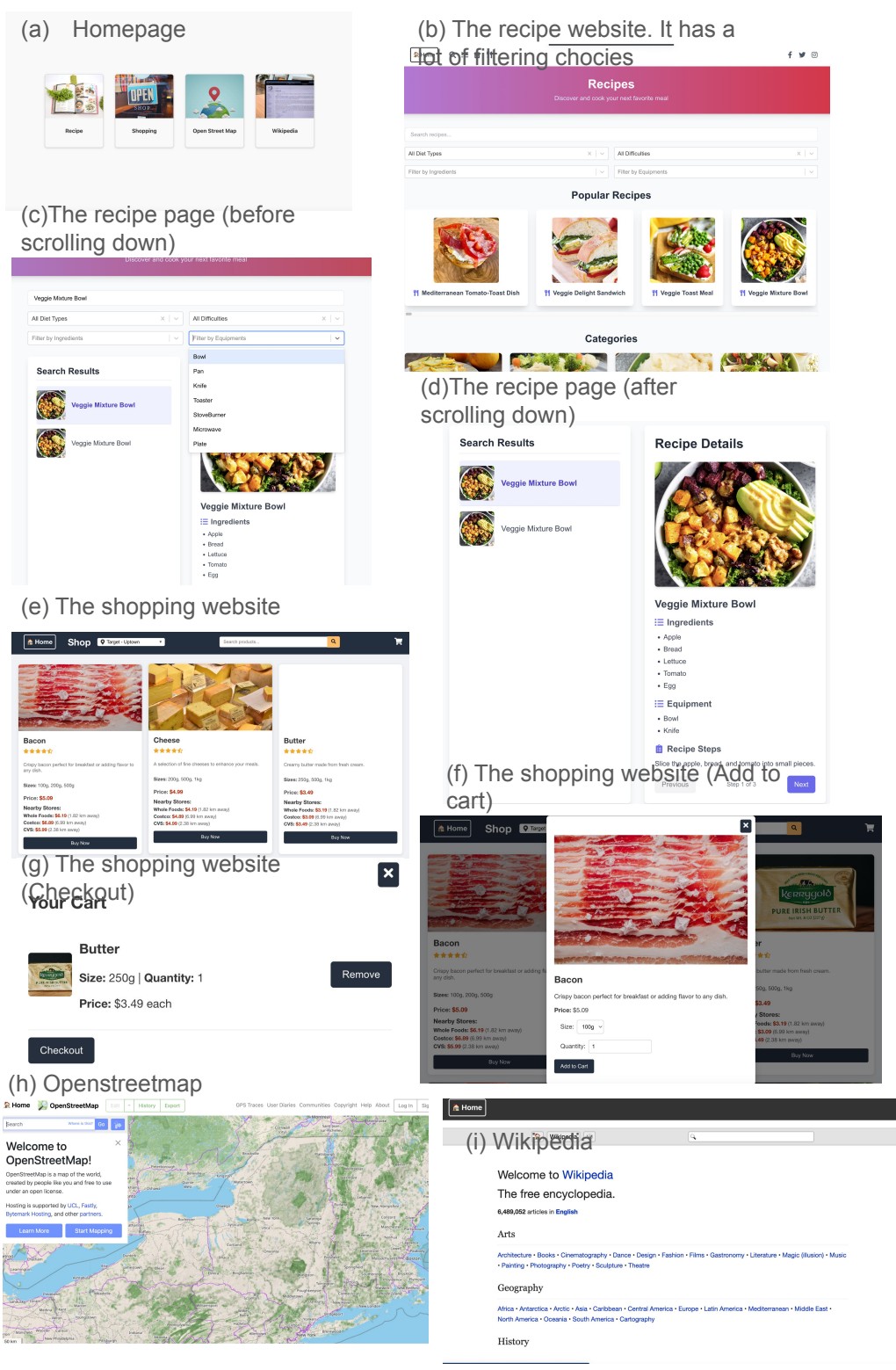

Figure 15: Web environment screenshots