# OpenReview forum: "Embodied Web Agents: Bridging Physical-Digital Realms for Integrated Agent Intelligence"
_NeurIPS.cc/2025/Datasets_and_Benchmarks_Track — NeurIPS 2025 Datasets and Benchmarks Track spotlight_

### Official Review · Reviewer_YrMv · 2025-06-29

**Rating:** 5
**Confidence:** 4

**Summary:**

The paper introduces "Embodied Web Agents," a simulation framework that combines internet information retrieval with embodied interaction, offering 3D environments and web interface interactions. It also constructs a benchmark covering a range of tasks such as cooking, navigation, shopping, traveling, and geolocation. After extensive testing of current large models, it reveals a significant gap between current AI capabilities and human abilities.

**Dataset Code Accessibility:**

Yes

**Dataset Code Comments:**

It provides access to the simulation environment including a web version, as well as detailed details of the baselines and provides reproduction code.

**Ethical Comments:**

This article is based on existing open source data and does not exist ethical concerns.

**Ethical Considerations:**

No, there are no or only very minor ethics concerns

**Final Justification:**

The author's reply resolved my doubts. This work is indeed a relatively large amount of work and makes a sufficient contribution to the community. I will raise my score to Accept.

**Limitations Weaknesses:**

1. Lack of Innovation: The paper appears to be more of an integration of existing components rather than introducing novel methodologies. For instance, it could focus on how to utilize existing benchmarks to collect accurate data and how to design precise planning methods that enable the correct completion of tasks. This approach would provide a clearer pathway for advancing the field rather than simply combining existing elements. Or simply putting forward your own insights, such as how to design a more suitable model or how to design a suitable reasoning process, I think it is also meaningful.
2. Absence of Physical Interaction: Despite the claim of embodied interaction, the actual interaction is merely a simple API call to execute actions, rather than physical control. While I understand that combining complex planning and physical interaction is challenging, the lack of physical interaction makes the paper seem like it's just adding knowledge base retrieval to planning tasks like navigation. This diminishes the paper's contribution. It would be much more significant if the paper included physical interaction and demonstrated how additional internet knowledge could improve the accuracy of physical execution.
3. Error Analysis: I am curious about the fact that the agent slices lettuce when the instruction was "slice apple", because this seems to be a low-level error that cannot be explained by cross-domain integration failures. Can you provide more visualization examples and analysis?

**Strengths Contributions:**

1. Clear Expression: The paper is well-written and easy to understand, with clear charts and diagrams that effectively illustrate the concepts.
2. Interesting Concept: The integration of information retrieval and embodied interaction is an intriguing idea. The tasks designed to highlight the importance of these two capabilities are relevant to real-world applications.
3. Diverse Benchmark: The EMBODIED WEB AGENTS Benchmark encompasses a diverse suite of tasks across multiple domains, including navigation, shopping, traveling, cooking, and geolocation. This benchmark is comprehensive and systematically tests an agent’s ability to bridge embodied perception, action, and web-based reasoning.

---

> ### Author Rebuttal · Authors · 2025-07-31
>
> *We appreciate the constructive comments from you, which are crucial for improving the paper! We have carried out additional experiments to resolve your concerns.*
>
> &nbsp;
>
> > **W1: Lack of Innovation: The paper appears to be more of an integration of existing components rather than introducing novel methodologies. For instance, it could focus on how to utilize existing benchmarks to collect accurate data and how to design precise planning methods that enable the correct completion of tasks. This approach would provide a clearer pathway for advancing the field rather than simply combining existing elements. Or simply putting forward your own insights, such as how to design a more suitable model or how to design a suitable reasoning process, I think it is also meaningful.**
>
> Thank you for raising your concern!
> * **Novel Paradigm Introduction** We introduce Embodied Web Agents as an entirely new class of AI systems that fluidly operate across both physical and digital realms. This is not simply combining existing elements, but establishing a fundamentally new research direction that addresses real-world scenarios where humans naturally integrate embodied and web-based reasoning. Integrating and utilizing web information for real-world tasks is very common, yet no existing papers or efforts have paid attention to this field. This already introduces a lot of novelty as a *benchmark paper*.
> * **Exhaustive Benchmark Construction**
> The creation of our benchmark involved unprecedented engineering efforts far beyond existing work. While the reviewer claims that we utilize existing benchmarks, we want to emphasize that *No prior benchmark exists for cross-domain embodied-web intelligence*. Constructing such benchmark required: (1) Complete web environment development: We engineered fully functional websites with React.js frontends and FastAPI backends. (2) Comprehensive data curation: We constructed meticulously designed tasks across five domains, each requiring extensive human verification, manual annotation, and quality assurance. (3) Novel integration architecture: We developed entirely new technical infrastructure to enable seamless switching between 3D embodied environments and web interfaces. (4) Custom tooling development: We built specialized annotation interfaces (Figure 8) with interactive map visualizations and trajectory analysis capabilities. (5) Integrating and bridging embodied and web environments is far from trivial, as it requires aligning fundamentally different modalities, interaction patterns, and temporal dynamics into a unified framework.
>
> This represents months of intensive engineering work creating infrastructure, environments, and evaluation protocols that enable systematic study of a previously unexplored research area. The technical complexity of maintaining task coherence across fundamentally different interaction modalities while ensuring reproducible evaluation cannot be understated.
> * **Methodological Insights**
> Our work highlights a key challenge: agents struggle not with individual skills, but with integrating reasoning across modalities. Effective coordination needs structured decision-making that unifies diverse modalities under common goals. Future embodied web agents will need hybrid reasoning systems to flexibly operate across physical and digital spaces.
> * **New Experiments & Insights**
> As a benchmark paper, our main goal is to provide a standardized evaluation for LLM-based agents, though here we are happy to include additional experiments and design insights.
>
> Based on our analysis above and inspired by W3 of Reviwer GnhX, we design several hybrid systems that integrate structured planning and decision making with LLMs. Specifically, We explore three hybrid approaches that integrate LLMs with structured planning to address cross-domain reasoning. GPT-HTN leverages hierarchical decomposition, combining GPT-4’s goal understanding with HTN planners. The Neuro-Symbolic model uses VLMs to extract symbolic representations from embodied contexts, enabling planning within the LLM framework. The Multi-Agent Planner distributes reasoning across domain-specialized LLMs coordinated by a central Director, facilitating modular control and persistent memory across environments.
>
> ||GPT|Gemini |GPT-HTN|Neuro-Symbolic|Multi-Agent LLM|
> |-|-|-|-|-|-|
> |Overall Acc|5.4|4.1|5.6|3.2|5.8|
> |Completion Rate|40.26|35.62|43.35|20.21|42.43|
> |Web Acc|59.71|47.74|47.59|22.58|49.92|
> |Embodied Acc|9.7|6.1|8.1| 4.4|6.0|
>
> The results reveal key differences in architectural strategies for cross-domain reasoning. GPT-HTN consistently outperforms the Neuro-Symbolic model, showing that hierarchical decomposition is more effective than flat LLM planning. The Multi-Agent LLM achieves the best overall performance. Its modular design and domain-specialized agents managed by a Director LLM enables flexible adaptation, persistent memory, and efficient task coordination.
>
> We also attach the results of training-based LLMs, as inspired by Reviewer UKAL.
>
> ||QWen2.5VL-7B|INTERNVL3-8B|3D-LLM|QWen(zero-shot)|INTERN(zero-shot)|
> |:-:|:-:|:-:|:-:|:-:|:-:|
> |Overall Acc|1.5|0.7|1.3|0.6|0.0|
> |Completion Rate|20.35|15.50|18.83|15.91|9.73|
> |Web Acc|43.57|30.82|28.56|28.65|10.64|
> |Embodied Acc|4.1|2.9|6.7|2.2|0.9|
>
> Please also refer to W2 of Reviewer UKAL for adaption-based methods results. We can see that despite using specialized training/adaption-based models, performance across all approaches remains unsatisfactory, with only slight improvements over zero-shot baselines, highlighting that the real challenge lies in integrating physical and digital reasoning.
>
> These results suggest that cross-domain reasoning cannot be solved by scale alone. It demands structured systems combining hierarchical planning, modular expertise, and centralized control. We hope these insights can inspire future researchers to develop new methods, and address your concerns to some extent.
>
> &nbsp;
>
> > **W2: Absence of Physical Interaction: Despite the claim of embodied interaction, the actual interaction is merely a simple API call to execute actions, rather than physical control. It would be much more significant if the paper included physical interaction and demonstrated how additional internet knowledge could improve the accuracy of physical execution.**
>
> We sincerely thank the reviewer for this important feedback and acknowledge that physical control is a crucial aspect of embodied AI systems.
> * **We Do Involve Physical Interaction** Our work does incorporate physical interaction through realistic simulated environments. In AI2-THOR, agents physically manipulate objects (picking up, slicing, cooking ingredients), navigate 3D spaces, and observe state changes through embodied perception. In our outdoor environments, agents physically traverse real-world street networks, make spatial decisions based on visual observations, and ground digital map instructions in physical navigation.
> * **Focus on High-Level Planning vs. Low-Level Control** We acknowledge that physical interaction can be decomposed into high-level planning and low-level control. Our work focuses on the former while utilizing pre-defined actions for low-level control. We want to emphasize that low-level control presents significantly greater complexity than high-level planning, particularly across our complex task domains. For instance, developing comprehensive low-level control policies for cooking alone would involve intricate motor coordination, force control, and object manipulation that could constitute an entire new research paper. To maintain focus on cross-domain integration, we deliberately isolate high-level planning, which is orthogonal to low-level control; comprehensive low-level policies can be independently developed and later integrated into our framework to replace pre-defined polocies. This also ensures reproducibility and allows a clean evaluation of agents’ reasoning abilities without low-level control confounds.
> * **Standard Practice**: This approach follows well-established precedent in highly-cited embodied AI benchmarks like ALFRED and BEHAVIOR, which focus on high-level planning using simulated environments. Similarly, major web agent benchmarks like WebArena and VisualWebArena use API-based interactions for studying high-level reasoning capabilities.
> * **New Experiments that Integrate Low-Level Control**
> To demonstrate our point, we finetune a VLA model which takes image observations as inputs and outputs low-level actions to interact with the ManipulaThor environment. We show the results below.
>
> |VLA|Overall Acc|Completion Rate|Web Acc|Embodied Acc|
> |-|-|-|-|-|
> ||0.0 |2.4|7.8|0.0|
>
> We can see that the model barely executes any task with low-level control successfully.
>
> &nbsp;
>
> > **W3: Error Analysis: I am curious about the fact that the agent slices lettuce when the instruction was "slice apple". Can you provide more visualization examples and analysis?**
>
> We apologize for not explaining this more clearly earlier. The reason the agent slices lettuce despite the instruction being "slice apple" is that the model hallucinates aspects of the recipe. Instead of adhering to the actual content of the webpage, the model generates its own imagined version of the dish. This reflects a common issue where the model fails to ground its actions in the given web context and instead relies on its own prior knowledge or assumptions about how certain dishes are typically made.
> You can find more visualization examples in the supplementary material. We are sorry that we cannot provide more due to rebuttal policy.
>
> &nbsp;
>
> *We wish that our response has addressed your concerns and turns your assessment to the positive side. We would really appreciate it if you could consider **Raising your score**. If you have more questions, feel free to let us know during the rebuttal window. We are willing to conduct any more experiments based on your requirements. Thank you!*
>
> &nbsp;
>
> Best,
>
> Authors

---

> > ### Comment · Area_Chair_HAmL · 2025-08-04
> > **Please engage with the authors' rebuttal**
> >
> > Hi Reviewer YrMv,
> >
> > The authors have provided a rebuttal addressing your points. Since there are only a couple days remaining for the discussion period with the authors, please engage with their rebuttal as soon as possible.
> >
> > Cheers,
> >
> > Your AC

---

> > ### Author Response · Authors · 2025-08-04
> > **Follow-Up on Rebuttal**
> >
> > Dear Reviewer,
> >
> > Thanks again for your suggestions to strengthen this work! As the rebuttal period is approaching the end soon, we want to know if our response has answered your questions and addressed your concerns. If no, we are more than happy to provide further modifications. If yes, would you kindly consider raising the score?
> >
> > Thanks again for your truly constructive and insightful feedback.
> >
> > Best, Authors

---

> > ### Comment · Reviewer_YrMv · 2025-08-05
> >
> > The author's reply resolved my doubts. This work is indeed a relatively large amount of work and makes a sufficient contribution to the community. I will raise my score to Accept.

---

> > > ### Author Response · Authors · 2025-08-05
> > > **Thank you!**
> > >
> > > Dear Reviewer YrMv,
> > >
> > > Thank you so much for your kind words and generous evaluation. We’re truly grateful that our response helped clarify your doubts. Your recognition of the effort and contribution means a great deal to us, and we deeply appreciate your decision to raise the score to Accept. Thank you again for your thoughtful review and support!
> > >
> > > Best,
> > > Authors

---

### Official Review · Reviewer_UKAL · 2025-06-30

**Rating:** 5
**Confidence:** 4

**Summary:**

This paper developed EMBODIED WEB AGENTS, a unified simulation platform that integrates indoor/outdoor 3D realistic environments with functional web interfaces. The authors also constructed and released the EMBODIED WEB AGENTS Benchmark, which consists of various tasks such as cooking, navigation, and geolocation. Their experiments revealed the performance gap between SOTA AI systems and human capabilities. Overall, the proposed method and benchmark are novel and interesting, but it would be better if the authors could:
1. Add more experiments and insightful analysis on the results.
2. Consider proposing or testing any learning methods to adapt or improve performance.

**Additional Feedback:**

Please see the weaknesses and limitations.

**Dataset Code Accessibility:**

Yes

**Dataset Code Comments:**

The URL and code are provided and checked.

**Ethical Comments:**

The authors use existing web data and environments, which have no ethical issues.

**Ethical Considerations:**

No, there are no or only very minor ethics concerns

**Final Justification:**

The authors have addressed my previous concerns. I increase my rating to Accept.

**Limitations Weaknesses:**

1. The environment design is asymmetric: outdoor tasks use real-world data, while indoor tasks rely solely on simulation. This limits the benchmark’s ability to evaluate cross-domain generalization.
2. The benchmark only tests frozen LLM-based agents in zero-shot settings. No training or adaptation methods are proposed or evaluated, which restricts its utility for studying agent improvement.
3. Evaluation focuses on structured metrics without human assessment of reasoning quality, interaction naturalness, or long-horizon planning, limiting the interpretability of agent performance.

**Strengths Contributions:**

Strengths:
1. The proposed paradigm is novel. It integrates physical action and web reasoning.
2. The benchmark is comprehensive and tasks are diverse, covering a wide range of tasks in real life.
3. The empirical insight on cross-domain integration failures is valuable.

Contributions:
1. The authors introduced EMBODIED WEB AGENTS as a new conceptual paradigm of AI systems that unify physical embodiment with web-scale knowledge access.
2. The authors provided a unified simulation platform that integrates indoor/outdoor 3D realistic environments with functional web interfaces, enabling agents to perform various cross-domain tasks.
3. The authors construct and release the EMBODIED WEB AGENTS Benchmark, including tasks like navigation, cooking and geolocation.
4. The authors conducted empirical analysis of SOTA LLMs on the proposed benchmark.

---

> ### Author Rebuttal · Authors · 2025-07-31
>
> *Thank you for your thorough review and the positive rating of our work! We greatly appreciate your insightful and constructive questions that have helped us reflect on and clarify important aspects of our approach.*
>
> &nbsp;
>
> > **W1: The environment design is asymmetric: outdoor tasks use real-world data, while indoor tasks rely solely on simulation. This limits the benchmark’s ability to evaluate cross-domain generalization.**
>
> We appreciate the reviewer's concern and acknowledge that the indoor and outdoor platforms have different levels of realism. However, we emphasize that each platform represents the optimal choice for our proposed tasks, and these varying realism levels are orthogonal to the core cognitive abilities we aim to evaluate.
> * **Optimal Platform Selection** The indoor and outdoor platforms represent the best available choices for each domain. For indoor tasks, AI2-THOR is the only platform that supports the complex object manipulation, state tracking, and cooking interactions our benchmark requires. No real-world indoor data collection can provide the precise controllability and comprehensive object states needed for systematic evaluation. For outdoor tasks, Google Earth represents the best available outdoor data source, offering comprehensive coverage across multiple cities with the navigation graph structure necessary for embodied movement. Importantly, both environments are specifically designed to integrate seamlessly with our websites and web-based information: AI2-THOR's object states and cooking progression can be directly mapped to recipe instructions from our web interfaces, while Google's geographic coordinates and visual landmarks naturally align with map services (e.g., Openstreetmap) and location-based web content (e.g., Wikipedia).
> * **Focus on Cross-Domain Capabilities:**
> Our benchmark is designed to evaluate cross-domain reasoning capabilities, where scene realism is largely orthogonal to the core cognitive challenges. The fundamental abilities we assess - grounding web instructions in embodied observations, maintaining cross-domain state representations, and dynamic environment switching, can be effectively evaluated regardless of the specific level of environmental realism. Although the two environments are of different realism lavels, both environments capture genuine real-world complexity that support the evaluation of the above capabilities. AI2-THOR uses photorealistic indoor scenes based on real home layouts, while Google provides authentic urban environments. Future improvements in scene realism (such as more realistic indoor data collection or enhanced cooking simulation fidelity) can be seamlessly incorporated into our framework without fundamentally altering the benchmark's core architecture, evaluation metrics, or task design principles.
> * **Cross-domain Generalization** Regarding cross-domain generalization, our benchmark actually enhances rather than limits generalization evaluation. The environmental asymmetry provides a more robust test of agents' adaptability. Successful cross-domain reasoning must work across different levels of environmental realism, from highly controlled simulations to noisy real-world data. Agents that can integrate web information with both photorealistic indoor simulations and authentic outdoor imagery demonstrate stronger generalization capabilities than those tested in uniform environments. This diversity better reflects real-world deployment scenarios where AI systems must operate across varied environmental conditions while maintaining consistent cross-domain reasoning abilities.
>
> &nbsp;
>
> > **W2: The benchmark only tests frozen LLM-based agents in zero-shot settings. No training or adaptation methods are proposed or evaluated, which restricts its utility for studying agent improvement.**
>
> Thank you so much for pointing this out! To address your concern, we attach training/adaption-based results below, and also validate our decision to use frozen LLM-based agents.
> * **Zero-Shot Methods** We use frozen LLM-based agents to establish clear baseline performance and identify fundamental challenges in cross-domain reasoning without confounding factors from task-specific training. This approach enables fair comparison across different model architectures and provides essential infrastructure for future research on agent improvement and adaptation methods.
> * **Training-Based Methods** However, we do find your suggestions valuable and conduct some experiments on trained LLMs. Specifically, we further generate 1,500 training data on cooking tasks. We train on these collected data and evaluate on our proposed benchmark. We use the open-source QWen2.5VL-7B model and INTERNVL3-8B model. We also include the 3D-LLM model which can perform planning in embodied environments.
>
> ||QWen2.5VL-7B|INTERNVL3-8B|3D-LLM|QWen(zero-shot)|INTERN(zero-shot)|
> |:-:|:-:|:-:|:-:|:-:|:-:|
> |**Overall Acc**|1.5|0.7|1.3|0.6|0.0|
> |**Completion Rate**|20.35|15.50|18.83|15.91|9.73|
> |**Web Acc**|43.57|30.82|28.56|28.65|10.64|
> |**Embodied Acc**|4.1|2.9|6.7|2.2|0.9|
>
> Despite the inclusion of training-based specialized models, performance remains unsatisfactory across all approaches, with training-based methods showing only marginal improvements over zero-shot baselines. Even models specifically trained for embodied tasks achieve modest gains in embodied reasoning, while maintaining varied web performance capabilities. This suggests that cross-domain switching remains fundamentally challenging regardless of training paradigm. The core difficulty lies not in individual domain competence but in the seamless integration and coordination between physical and digital reasoning that our benchmark uniquely evaluates.
>
> * **Adaption-Based Methods** We further conduct experiments on adaption-based methods for outdoor settings and include the results below.
>
> In the table below, we evaluate four settings: zero‑shot, one‑shot, three‑shot, and five‑shot. In the paper, we report the zero‑shot results. As the number of shots increases, performance steadily improves, but the marginal gains shrink with more additional shots. Moreover, even with these few‑shot gains, it remains very difficult for VLMs to handle our cross‑domain complex tasks.
>
> |||GPT(zero-shot)|GPT(one-shot)|GPT(three-shot)|GPT(five-shot)|
> |:-:|:-:|:-:|:-:|:-:|:-:|
> |**Navigation**|**Overall Acc**|34.72|35.42|36.11|36.81|
> ||**Completion Rate**|52.08|52.78|53.47|54.01|
> ||**Web Acc**|69.44|70.14|70.83|71.32|
> ||**Embodied Acc**|48.61|49.36|50.03|50.61|
> |**Shopping**|**Overall Acc**|25.46|25.93|26.35|26.78|
> ||**Completion Rate**|31.94|32.46|32.89|33.32|
> ||**Web Acc**|39.35|39.81|40.25|40.70|
> ||**Embodied Acc**|34.26|34.75|35.15|35.61|
> |**Traveling**| **Overall Acc**|30.91|31.82|32.74|33.64|
> ||**Completion Rate**|50.91|51.86|52.73|53.63|
> ||**Web Acc**|57.27|58.23|59.12|59.93|
> ||**Embodied Acc**|47.27|48.20|49.11|49.88|
>
> Based on the GPT few-shot experiments, we ultimately chose the three-shot setting and applied it to all the models we evaluated. Below are the results for each model under the three-shot setting, which can be compared against Table 2 in our paper. We observe that, with three shots, all models achieve relatively better performance. Even so, there remains a large gap compared to human performance on our proposed benchmark. Both (1) few-shot experiments across different models and (2) experiments with varying shots on the same model show that few-shot samples can improve performance to some degree, but the gains are limited. Moreover, all VLMs still lag far behind humans, revealing their current inability to handle complex cross-domain (embodied‑web) tasks and underscoring the challenge posed by our benchmark, as well as the insights and future development directions it offers for the research field.
>
>
> |||GPT(three-shot)|Gemini(three-shot)|Qwen(three-shot)|Intern(three-shot)|Human|
> | :-: | :-: | :-: | :-: | :-: | :-: | :-: |
> |**Navigation**|**Overall Acc**|36.11| 31.94 |16.69|13.91|90.28|
> ||**Completion Rate**|53.47|50.35|38.19|27.43|91.32|
> ||**Web Acc**|70.83|68.75|58.33|40.28|92.36|
> ||**Embodied Acc**|50.03|37.92|32.64 |24.41|90.97|
> |**Shopping**|**Overall Acc**|26.35|24.54|14.81|11.62|92.5 |
> ||**Completion Rate**|32.89|31.48|19.91|15.28|93.52|
> ||**Web Acc**|40.25|38.51|24.58|18.06|93.06 |
> ||**Embodied Acc**|35.15 |35.19|18.98|14.35 |93.98 |
> |**Traveling**|**Overall Acc**|32.74|27.27|13.69|10.93|91.82 |
> ||**Completion Rate**|52.73|49.09|37.27| 22.76|93.64 |
> ||**Web Acc**|59.12|54.55|43.68|27.35 |94.55 |
> ||**Embodied Acc**|49.11|46.36|30.91|20.98|92.73 |
>
>
> * We want to emphasize that our data pipeline and engine are fully open-sourced. Future researchers can leverage the data pipeline to generate training data as well.
>
> &nbsp;
>
> > **W3: Evaluation focuses on structured metrics without human assessment of reasoning quality, interaction naturalness, or long-horizon planning, limiting the interpretability of agent performance.**
>
> Thank you for the suggestion! Human evaluation is indeed very important and we attach the results below and will incorporate them into the main paper for camera-ready version.
>
> || GPT | Gemini | Qwen| INTERN|
> |:-:|:-:|:-:|:--:|:-:|
> |**Reasoning Quality** | 4.0| 3.5| 3.2| 2.5|
> |**Interaction Naturalness** | 4.4| 4.8| 4.0|4.2|
> |**Long-horizon Planning** | 3.7| 3.3  | 2.4| 1.7|
> |**Overall**| 4.2|4.0| 3.5| 3.1 |
>
> &nbsp;
>
> *Thank you again for your thoughtful questions that have helped us clarify key aspects of our work! We hope our detailed responses have addressed your concerns and further demonstrated the strengths of our approach, and therefore have reinforced your confidence in our work. If so, would you kindly consider **raising your score**? Thank you! We will incorporate these clarifications in the final version to better present our contributions.  If you have any additional questions, we would be happy to address them during the rebuttal window. Thanks again!*
>
> &nbsp;
>
> Best,
>
> Authors

---

> > ### Comment · Area_Chair_HAmL · 2025-08-04
> > **Please engage with the authors' rebuttal**
> >
> > Hi Reviewer UKAL,
> >
> > The authors have provided a rebuttal addressing your points. Since there are only a couple days remaining for the discussion period with the authors, please engage with their rebuttal as soon as possible.
> >
> > Cheers,
> >
> > Your AC

---

> > ### Comment · Reviewer_UKAL · 2025-08-04
> >
> > The authors have addressed my previous concerns. I increase my rating to Accept.

---

> > > ### Author Response · Authors · 2025-08-04
> > > **Thank you for your review!**
> > >
> > > Dear Reviewer UKAL,
> > >
> > > Thank you for your timely and constructive feedback, which is truly encouraging. We deeply appreciate your thoughtful reviews and valuable comments—they have been instrumental in strengthening our paper. It was a pleasure engaging in this discussion, and we are glad that we could address your concerns.
> > >
> > > Best regards,
> > >
> > > Authors

---

### Official Review · Reviewer_GnhX · 2025-07-02

**Rating:** 5
**Confidence:** 5

**Summary:**

The paper introduces a novel paradigm for AI agents that unify physical embodiment and web-scale reasoning. The authors propose a simulation platform, which integrates realistic 3D indoor and outdoor environments with interactive web interfaces, enabling agents to fluidly operate across both physical and digital domains. They introduce a comprehensive benchmark comprising ~1.5k tasks spanning navigation, cooking, shopping, traveling, and geolocation, all requiring cross-domain intelligence. Experiments using state-of-the-art LLM agents reveal significant performance gaps compared to human capabilities, particularly in cross-domain integration. The work highlights new challenges in building truly integrated intelligent agents and offers an open-source benchmark to facilitate future research.

**Dataset Code Accessibility:**

Yes

**Ethical Considerations:**

No, there are no or only very minor ethics concerns

**Final Justification:**

I found that the dataset documents are not very useful. The benchmark brings together a lot of static datasets and web interfaces, but it is not clear how to use it properly. Also, there is no guide about how to control embodied agents in the Unreal Engine, as shown in the paper.

I have raised my score to accept based on the AC's advice. Nevertheless, to make it a useful benchmark, providing an intuitive guide is necessary. I hope the authors could continue improving this benchmark to make it really useful.

**Limitations Weaknesses:**

- The benchmark is built entirely in simulation (Sec. 3), which may limit its applicability to real-world agents due to the gap between simulated and real-world perception and interaction.

- Although the paper introduces diverse tasks, the coverage of some domains (e.g., tourism and geolocation) remains limited in scale compared to cooking (911 tasks vs. 110 and 142; Sec. 4).

- The evaluation focuses solely on LLM-based agents without exploring specialized planners or hybrid models that might better handle cross-domain reasoning (Sec. 5.1).

- The benchmark’s dependency on specific web environments (e.g., custom recipe and shopping sites; Sec. 3.3) may constrain generalization to open web scenarios not covered in the platform.

The benchmark is built on top of many existing simulation platforms, and their technical contribution appears to be merely assembling them together.

**Strengths Contributions:**

- The paper introduces a novel benchmark and simulation platform that uniquely integrates embodied environments with web-scale interfaces to evaluate cross-domain intelligence.

- It provides a large, realistic, and publicly available dataset spanning ~1.5k tasks across five domains, enabling rigorous evaluation of agents in both physical and digital settings.

- The empirical analysis is thorough, revealing significant performance gaps in state-of-the-art models and highlighting cross-domain reasoning as a key challenge, with clear visualizations and well-structured presentation.

---

> ### Author Rebuttal · Authors · 2025-07-31
>
> *Thank you for your thoughtful review! We are delighted to address your questions by providing additional insights and carrying out additional experiments.*
>
> &nbsp;
>
> > **W1: The benchmark is built entirely in simulation (Sec. 3), which may limit its applicability to real-world agents due to the gap between simulated and real-world perception and interaction.**
>
> We greatly appreciate your concern and offer several perspectives to justify our design choices.
> * **Real-World Complexity Introduces Orthogonal Challenges** Deploying real-world agents would introduce substantial noise from low-level control and perception challenges. Developing real-world cooking robots or city-scale navigation agents are non-trivial, and could each constitute entire research papers in themselves. Our aim is to evaluate agents' cognitive capabilities to integrate embodied and web-based reasoning without the confounding factors of perceptual perturbations, hardware limitations or control instability. By using simulation, we isolate the core problem of cross-domain reasoning. The specific choice of embodied platform is orthogonal to our contribution; once real-world systems mature, our benchmark can be readily adapted without requiring fundamental architectural changes.
> * **Optimal Choices for Task Requirements** We emphasize that our chosen simulators represent the optimal available platforms for each domain's specific requirements. For indoor tasks, AI2-THOR is the only platform that supports the complex object manipulation, state tracking, and cooking interactions our benchmark requires. No current real-world setup can provide the precise controllability, comprehensive object state monitoring, and systematic evaluation capabilities needed for reproducible research at this scale. For outdoor tasks, Google Earth represents the most comprehensive real-world data source available, offering authentic urban environments across multiple cities with the navigation graph structure necessary for embodied movement. Both platforms are specifically designed to integrate seamlessly with our websites and web-based information, and serve perfectly for our embodied web agent tasks.
> * **Focus on Core Challenges** Our benchmark isolates the high-level reasoning abilities necessary for web-embodied intelligence, such as grounding web instructions in visual scenes, maintaining coherent state across modalities, and planning across digital and physical contexts. These challenges persist regardless of whether the agent operates in simulation or reality, as they stem from cognitive integration rather than sensorimotor fidelity. The poor performance of state-of-the-art models in our simulated setup already highlights the difficulty of these tasks, even before introducing real-world noise.
> * **Reproducibility and Controllability** Simulation environments provide essential reproducibility and controllability that real-world deployments cannot match. Our benchmark enables systematic evaluation across thousands of tasks with consistent environmental conditions, precise state monitoring, and deterministic evaluation metrics. Real-world experiments would introduce uncontrollable variables (weather, lighting, object positioning, hardware variability) that would make fair comparison across different agent architectures impossible.
>
> &nbsp;
>
> > **W2: Although the paper introduces diverse tasks, the coverage of some domains (e.g., tourism and geolocation) remains limited in scale compared to cooking (911 tasks vs. 110 and 142; Sec. 4).**
>
> Thank you for pointing this out!
> * **Indoor vs. Outdoor Task Balance** While there appears to be an imbalance when examining individual domains, the overall distribution between indoor (cooking: 911 tasks) and outdoor tasks (navigation: 144, shopping: 216, traveling: 110, geolocation: 142; total: 612 tasks) is reasonably balanced. The outdoor tasks collectively evaluate similar core abilities (e.g.,spatial reasoning, navigation, and grounding web-based geographic information in physical environments), across different contexts and complexity levels.
> * **Established Data Pipeline for Future Expansion** We have established and open-sourced robust data collection pipelines and annotation frameworks for all domains. Our infrastructure is designed to support easy expansion. We can and will include more examples in future releases. The current distribution provides sufficient coverage to demonstrate cross-domain reasoning capabilities and establish baseline performance across diverse task types.
>
> &nbsp;
>
> > **W3: The evaluation focuses solely on LLM-based agents without exploring specialized planners or hybrid models that might better handle cross-domain reasoning (Sec. 5.1).**
>
> Thank you for this suggestion! As a benchmark paper, our primary goal is to provide a standardized evaluation framework for LLM-based agents. While we intended to leave broader exploration of the benchmark to future researchers, we are more than willing to address your concern by proposing and investigating several hybrid methods that integrate traditional planners with LLMs. We list the methods and their performances below.
> * GPT-HTN: A hierarchical planning system that combines GPT-4 with Hierarchical Task Network architecture. The system uses LLMs to decompose high-level goals into hierarchical subtasks, then employs HTN planners for detailed action sequencing. The framework handles complex multi-step planning by leveraging GPT's natural language understanding for goal interpretation while using HTN's structured approach for logical task ordering and execution.
> * Neuro-Symoblic: An architecture where VLMs extract symbolic states and objects from embodied environments, then use LLMs to generate optimal action sequences from these symbolic representations.
> * Multi-Agent Planner: An architecture that uses multiple specialized LLMs in a compositional framework - a "Director LLM" for high-level cross-domain coordination, a "Web Specialist LLM" for web navigation, an "Embodied Specialist LLM" for spatial reasoning and object manipulation. Each specialist operates within its domain while the Director manages information flow and decision handoffs between domains through structured communication.
>
> We attach the results of indoor cooking task below.
> |      |          GPT | Gemini | **GPT-HTN** | **Neuro-Symbolic** | **Multi-Agent LLM** |
> |:-:|:-:|:-:|:-:|:-:|:-:|
> | **Overall Acc**    | 5.4 | 4.1 | 5.6         | 3.2                | 5.8                 |
> | **Completion Rate** | 40.26 | 35.62 | 43.35       | 20.21              | 42.43               |
> | **Web Acc**        | 59.71 | 47.74 | 47.59       | 22.58              | 49.92               |
> | **Embodied Acc**   | 9.7 | 6.1 | 8.1         | 4.4                | 6.0                 |
>
> The results indicate that hierarchical approaches like GPT-HTN perform better than sequential models (baseline LLMs as in our paper), highlighting the importance of structured task decomposition. The superior performance of the Multi-Agent LLM indicates that to effectively handle cross-domain (embodied-web) reasoning, a high-level director or centralized controller is essential.
>
> &nbsp;
>
> > **W4: The benchmark’s dependency on specific web environments (e.g., custom recipe and shopping sites; Sec. 3.3) may constrain generalization to open web scenarios not covered in the platform.**
>
> Thank you for the question.
>  * **Real-World Website Complexity & Instability**: Real-world websites often contain content, procedures, or requirements that cannot be executed within our simulator environments (e.g.,specialized equipment not modeled in AI2-THOR, shops not covered in our Google data, products unavailable in our shopping catalog, or complex multi-step procedures not supported by our action spaces). These mismatches between web content and simulator capabilities would create systematic evaluation failures unrelated to cross-domain reasoning abilities. Moreover, real-world websites are constantly updating, therefore cannot provide reproducibility for our benchmark.
>  * **Representative Web Functionality** Our custom websites implement core web interaction patterns found across the internet: search interfaces, navigation menus, product catalogs, and information retrieval. These fundamental interaction modalities are sufficient to evaluate the core capabilities proposed by this paper.
> * **Future Extensibility** Our framework is designed to accommodate additional web environments or real-world websites. The modular architecture allows researchers to integrate new web interfaces without requiring fundamental changes to the benchmark infrastrcture.
>
> &nbsp;
>
> *We once again thank your efforts in reviewing our paper! Your valuable comments really help improve our paper, and inspire us a lot! We hope our response has addressed your concerns, and therefore you would consider **raising your score**. If you have more questions, don't hesitate to let us know in the rebuttal window! Thanks again!*
>
> &nbsp;
>
> Best,
>
> Authors

---

> > ### Comment · Reviewer_GnhX · 2025-08-02
> >
> > Thanks for the rebuttal, which addresses my concerns. My only concern is that the benchmark is built on top of many existing simulation platforms, and their technical contribution appears to be merely assembling them together. However, I do understand that combining all components also requires a significant amount of effort. Hence, I decided to keep my original rating as borderline accept.

---

> > > ### Comment · Area_Chair_HAmL · 2025-08-04
> > > **Clarification about your response**
> > >
> > > Hi Reviewer GnhX,
> > >
> > > Thanks for your review and responding to the authors. I am a bit curious about your final conclusion, since you acknowledge that the rebuttal addressed most of your concerns apart from the concern that the main contribution seems to be assembling many existing components.
> > >
> > > The more important question for D&B is whether this work constitutes a useful dataset or benchmark for the community (or a subpart of the community), and whether it is easy to use, well-documented and accessible, and whether it has sufficient detail about the procedure for collecting the datasets and creating the benchmark. I also do not fully understand if the rebuttal addresses several of your concerns why this wouldn't result in a score adjustment (please use Borderline sparingly).
> > >
> > > I am okay with you sticking to your score, but please provide more justification in line with the above points (also see the Reviewer guidelines).
> > >
> > > Thanks for your work in supporting Neurips.
> > >
> > > Cheers,
> > > Your AC

---

> > > ### Author Response · Authors · 2025-08-04
> > > **Thank you for your response & Further clarification**
> > >
> > > Dear Reviewer GnhX,
> > >
> > > *We sincerely appreciate your detailed reviews and insightful comments, which have played a key role in improving our paper. We’re pleased to have addressed your concerns.*
> > >
> > > However, we'd like to further clarify the novelty and technical contributions of this paper.
> > >
> > > * **Novel Paradigm Introduction** We introduce Embodied Web Agents as an entirely new class of AI systems that fluidly operate across both physical and digital realms. This is not simply combining existing elements, but establishing a fundamentally new research direction that addresses real-world scenarios where humans naturally integrate embodied and web-based reasoning. Integrating and utilizing web information for real-world tasks is very common and useful, yet no existing papers or efforts have paid attention to this field. This already introduces a lot of novelty as a *benchmark paper*.
> > > * **Exhaustive Benchmark Construction**
> > > The creation of our benchmark involved unprecedented engineering efforts far beyond existing work. This required: (1) Complete web environment development: We engineered fully functional websites with React.js frontends and FastAPI backends. (2) Comprehensive data curation: We constructed meticulously designed tasks across five domains, each requiring extensive human verification, manual annotation, and quality assurance. (3) Novel integration architecture: We developed entirely new technical infrastructure to enable seamless switching between 3D embodied environments and web interfaces. (4) Custom tooling development: We built specialized annotation interfaces (Figure 8) with interactive map visualizations and trajectory analysis capabilities. (5) Integrating and bridging embodied and web environments is far from trivial, as it requires aligning fundamentally different modalities, interaction patterns, and temporal dynamics into a unified framework.
> > >
> > > This represents months of intensive engineering work creating infrastructure, environments, and evaluation protocols that enable systematic study of a previously unexplored research area. The technical complexity of maintaining task coherence across fundamentally different interaction modalities while ensuring reproducible evaluation cannot be understated.
> > >
> > > * **Methods & New Experiments**
> > > We evaluate on SOTA LLM agents and highlights a key challenge: agents struggle not with individual skills, but with integrating reasoning across modalities. Effective coordination needs structured decision-making that unifies diverse modalities under common goals. To provide more insights about potential methods, we take the suggestions of all reviewers and carry out the experiments below:
> > >     * **[New experiments on three hybrid planning methods (Reviewer GnhX)]** Per your request, we design three methods that integrate classical planning with LLMs: GPT-HTN, Neuro-Symbolic and Multi-Agent LLMs. Results show that structural and hierarchical planning with a high-level commander is beneficial, providing insights for future model development on our benchmark.
> > >     * **[New experiments on training-based methods (Reviewer UKAL)]** We add training-based methods where we train on newly-collected data and evaluate on our benchmark. Results show that even with training, the models are still far from satisfying, demonstrating the challenges of our benchmarks.
> > >     * **[New experiments on adaption-based methods (Reviewer UKAL)]** We also add few-shot experiments and play with different number of shots on different methods. This only shows minor improvements over zero shot.
> > >     * **[New experiments with VLA models on low-level actions (Reviewer YrMv)]** We experiment with VLA models to output low-level embodied actions, which show near-zero performances, suggesting that low-level actions introduce much noise to the problem.
> > >     * **[New experiments of different viewpoints (Reviewer PFwd)]** We experiment with using observations from first-person view and third-person view, and show that the model performs better with first-person view.
> > >
> > > We believe that these experiments together show a lot of insights from the methodology perspectives, besides all the non-trivial contributions we made in the benchmark construction.
> > >
> > > * **Accessibility** As response to the AC, our benchmark is easy to use, well-documented and accessible. We have open-sourced the codes of models, simulator environments, web environments as well as data construction in the github link: https://github.com/Embodied-Web-Agent/Embodied-Web-Agent. We have a webpage built for this benchmark https://embodied-web-agent.github.io/ and our dataset is hosted on the huggingface as well.
> > >
> > > *Thank you again for your thoughtful feedbacks, and all of your time and efforts reviewing our paper and reading our rebuttal!*

---

> > > > ### Comment · Reviewer_GnhX · 2025-08-05
> > > >
> > > > Dear authors,
> > > >
> > > > I found that the dataset documents are not very useful. The benchmark brings together a lot of static datasets and web interfaces, but it is not clear how to use it properly. Also, there is no guide about how to control embodied agents in the Unreal Engine, as shown in the paper.
> > > >
> > > > I have raised my score to accept based on the AC's advice. Nevertheless, to make it a useful benchmark, providing an intuitive guide is necessary. I hope the authors could continue improving this benchmark to make it really useful.

---

> > ### Comment · Reviewer_GnhX · 2025-08-05
> >
> > Dear AC,
> >
> > Thanks for your response. I adjusted my score to **accept** based on your advice.
> >
> > Justifications:
> >
> > I found that the dataset documents are not very useful. The benchmark brings together a lot of static datasets and web interfaces, but it is not clear how to use it properly. Also, there is no guide about how to control embodied agents in the Unreal Engine, as shown in the paper. I hope the authors could continue improving this benchmark to make it really useful.

---

> > > ### Author Response · Authors · 2025-08-05
> > > **Thank you**
> > >
> > > Dear Reviewer GnhX,
> > >
> > > Thank you for your thoughtful consideration and for updating your score. We appreciate your support and are glad our response was helpful in clarifying our work.
> > >
> > > Best,
> > > Authors

---

### Official Review · Reviewer_PFwd · 2025-07-24

**Rating:** 5
**Confidence:** 4

**Summary:**

The authors proposed a novel paradigm, Embodied Web Agents, which incorporates 3D indoor and outdoor environments with functional web interfaces. They also proposed a novel platform and a benchmark that covers a range of tasks requiring both 3D reasoning and online information exploration, by marrying the indoor environment AI2-THOR, the outdoor environment Google Earth, as well as web interfaces like Wikipedia. Evaluation on state-of-the-art methods still lags greatly behind human performance, showing a critical aspect of current VLMs that needs improvement.

**Dataset Code Accessibility:**

Yes

**Ethical Considerations:**

No, there are no or only very minor ethics concerns

**Final Justification:**

After reading the rebuttal and reviews from other reviewers, I keep my original rating 'accept'.

**Limitations Weaknesses:**

- A big part of the task is exploring internet information, while related works (e.g., Hu, H., Luan, Y., Chen, Y., Khandelwal, U., Joshi, M., Lee, K., … & Chang, M. W. (2023). Open-domain visual entity recognition: Towards recognizing millions of Wikipedia entities) missed studies in this area. It is suggested to discuss them.
- It could be informative to provide first-person view, third-person view, and integration of both in this dataset to further examine whether there are viewpoint types favored by models.
- Although the proposed task is intended for VLM evaluation, it would still be interesting to see how well traditional embodied AI methods and their integration with internet recognition methods would perform in this task.

**Strengths Contributions:**

- The proposed task is innovative. By bridging embodied AI with internet knowledge, it would greatly expand the embodied AI’s ability to deal with real-world tasks. Also, the proposed platform and benchmark seem to be expandable. One can further enhance the benchmark to enable embodied AI in various tasks and provide insights into designing methods for real-world applications as well.
- The dataset is thorough and extensive, containing a platform for internet exploration and allowing 1.5K+ tasks, including navigation, shopping, traveling, geolocation, etc. This broad coverage enables in-depth analysis of methods.
- Various state-of-the-art VLMs are evaluated, and their weaknesses (such as cross-modal integration) are extensively analyzed, providing insights for model improvement.

---

> ### Author Rebuttal · Authors · 2025-07-31
>
> *We appreciate the positive and constructive comments from you, which are crucial for improving the paper! We have carried out additional experiments to resolve your concerns.*
>
> &nbsp;
>
> > **W1: A big part of the task is exploring internet information, while related works (e.g., Hu, H., Luan, Y., Chen, Y., Khandelwal, U., Joshi, M., Lee, K., … & Chang, M. W. (2023). Open-domain visual entity recognition: Towards recognizing millions of Wikipedia entities) missed studies in this area. It is suggested to discuss them.**
>
> Thank you so much for your suggestion! OVEN is indeed a very good paper which challenges models to link images to specific Wikipedia entities given text queries, and is highly relevant to our paper. We have revised our paper to include OVEN, which will be reflected in our camera-ready and preprint version.
>
> &nbsp;
>
> > **W2: It could be informative to provide first-person view, third-person view, and integration of both in this dataset to further examine whether there are viewpoint types favored by models.**
>
> Very good suggestion! We have included experimental results of models paired with first-person view observations, third-person view observations, and integration of both below.
>
>
> |         | **First-Person View** |   | **Third-Person View** |   | **Integration** |    |
> |:-------------------:|:---------------------:|:------:|:---------------------:|:------:|:---------------:|:------:|
> |                 | GPT                   | Gemini | GPT                   | Gemini | GPT             | Gemini |
> | **Overall Acc**     | 5.4                   | 4.1    | 4.9                   | 3.5    | 5.2             | 4.5    |
> | **Completion Rate** | 40.26                 | 35.62  | 38.82                 | 36.91  | 43.32           | 37.43  |
> | **Web Acc**         | 59.71                 | 47.74  | 59.90                 | 50.71  | 57.60           | 45.56  |
> | **Embodied Acc**    | 8.7                   | 6.1    | 7.9                   | 4.8    | 8.9             | 6.7    |
>
> The cooking performance results reveal that first-person view (FPV) consistently outperforms third-person view (TPV) across both models, though the differences are modest. GPT achieves 5.4% overall accuracy with FPV compared to 4.9% with TPV, while Gemini shows 4.1% versus 3.5% respectively. This suggests that the egocentric perspective provides slight advantages for embodied reasoning, likely due to better alignment between the agent's visual experience and action execution. The integration approach, which combines multiple viewpoints, shows the most balanced performance with GPT reaching 5.2% overall accuracy and improved completion rates. However, the relatively small performance variations across viewpoints indicate that while FPV offers marginal benefits, the fundamental challenges in cross-domain integration, coordinating web instructions with embodied actions, persist regardless of visual perspective, reinforcing that the core bottleneck lies in cognitive integration rather than perceptual representation.
>
> &nbsp;
>
> > **W3: Although the proposed task is intended for VLM evaluation, it would still be interesting to see how well traditional embodied AI methods and their integration with internet recognition methods would perform in this task.**
>
> Thank you for pointing this out! Inspired by this point as well as W3 by Reviewer GnhX, we propose several hybrid methods which integrate traditional embodied AI methods (e.g., hierarchical planning using HTN, neuro-symbolic methods for sense-think-act) with modern LLMs. We introduce the methods and attach the results below.
>
> * GPT-HTN: A modern extension of traditional Hierarchical Task Network (HTN) planning, where high-level goals are recursively decomposed into subtasks until reaching primitive actions. In classical robotics, this approach provided interpretable and logically consistent execution paths, but required extensive manual authoring of decomposition rules. GPT-HTN replaces that rigidity with the flexible abstraction capabilities of LLMs like GPT-4, which dynamically generate decompositions from natural language goals and environmental context. For embodied-web agents, goals like “make a cake” or “prepare for a trip” can be interpreted broadly across domains, and GPT-HTN allows the system to parse such goals into mixed-domain subtasks like “buy ingredients online” and “place items in the fridge.” The LLM handles language ambiguity and domain translation, while the HTN planner ensures that task execution is logically structured and recoverable.
> * Neuro-Symoblic: Contemporary embodiments of the Sense-Think-Act paradigm. Traditionally, this model separated perception, cognition, and control into discrete stages, with perception modules converting sensor input into symbolic world models, which were then used by symbolic planners. In our version, perception is handled by neural modules like vision-language models (VLMs), which extract structured symbolic representations from embodied environments. The planning phase is handled by an LLM which performs symbolic reasoning over these extracted representations.
> * Multi-Agent Planner: This method draws directly from traditional modular reactive architectures, such as Brooks’ subsumption architecture or hybrid layered control systems. These classic systems assign specific modules to handle different behaviors (e.g., locomotion, obstacle avoidance, memory), often operating concurrently with arbitration mechanisms. In the Multi-Agent Planner, this modularity is reinterpreted as a system of specialized LLMs, each fine-tuned for a particular domain: a “Web Specialist” LLM for navigating digital interfaces, an “Embodied Specialist” for spatial reasoning and object manipulation, and a “Director” LLM that conducts their interaction.
>
> The results of indoor cooking are shown below.
>
> |         |          GPT | Gemini | **GPT-HTN** | **Neuro-Symbolic** | **Multi-Agent LLM** |
> |:-:|:-:|:-:|:-:|:-:|:-:|
> | **Overall Acc**    | 5.4 | 4.1 | 5.6         | 3.2                | 5.8                 |
> | **Completion Rate** | 40.26 | 35.62 | 43.35       | 20.21              | 42.43               |
> | **Web Acc**        | 59.71 | 47.74 | 47.59       | 22.58              | 49.92               |
> | **Embodied Acc**   | 9.7 | 6.1 | 8.1         | 4.4                | 6.0                 |
>
> Our preliminary experiments with hybrid approaches show modest improvements over baselines. This shows that integrating traditional embodied AI methods (e.g., SLAM navigation, behavior trees, and classical planners) with modern web recognition capabilities represents a promising avenue for addressing the fundamental cross-domain integration challenges our benchmark reveals. Thank you again for giving us this valuable suggestion!
>
> &nbsp;
>
> *We believe these comprehensive revisions and additional experiments have substantially strengthened our paper and addressed your thoughtful comments. If you have further questions, please don't hesitate to let us know in the rebuttal window! It'd be great if you could also consider **raising your score** as well. Thank you again for your invaluable feedback in helping us enhance this paper!*
>
> &nbsp;
>
> Best,
>
>
> Authors

---

> > ### Comment · Area_Chair_HAmL · 2025-08-04
> > **Please engage with the authors' rebuttal**
> >
> > Hi Reviewer PFwd,
> >
> > The authors have provided a rebuttal addressing your points. Since there are only a couple days remaining for the discussion period with the authors, please engage with their rebuttal as soon as possible.
> >
> > Cheers,
> > Your AC

---

> > ### Comment · Reviewer_PFwd · 2025-08-05
> > **The author's response addressed my concerns.**
> >
> > Thanks for the rebuttal. The authors addressed all my previous concerns, and I will keep my original rating accept.

---

> > > ### Author Response · Authors · 2025-08-05
> > > **Thank you**
> > >
> > > Dear Reviewer PFwd,
> > >
> > > Thank you for your follow-up and for taking the time to review our rebuttal. We're glad to hear that our responses addressed your concerns, and we truly appreciate your decision to maintain your acceptance recommendation.
> > >
> > > Best,
> > > Authors

---

### Note · Authors · 2025-08-12

*We genuinely thank all reviewers and ACs for their efforts and time in reviewing our paper, as well as their constructive suggestions! We sincerely appreciate **the positive 5-5-5-5** evaluation from PFwd, GnhX, UKAL, and YrMv.*

# Contributions
* **Novel Paradigm Introduction** We introduce Embodied Web Agents as a new class of AI systems that bridge physical and digital realms. *Proposed task is innovative* (PFwd); *Novel paradigm* (UKAL); *Uniquely integrates embodied environments with web-scale interfaces* (GnhX); *Interesting concept* (YrMv).
* **Technical Development** We built a unified platform integrating AI2THOR, Google Earth, and web tools with frontends, backends, and annotation systems. *Platform and benchmark seem expandable* (PFwd); *Unified simulation platform that integrates indoor/outdoor 3D realistic environments with functional web interfaces* (UKAL); *Systematically tests an agent's ability to bridge embodied perception, action, and web-based reasoning* (YrMv)
* **Comprehensive Benchmark** We created a large-scale benchmark spanning five domains-cooking, navigation, shopping, tourism, and geolocation. *Thorough and extensive* (PFwd); *Large, realistic, and publicly available* (GnhX); *comprehensive and diverse, covering a wide range of tasks across multiple domains in real life* (UKAL, YrMv).
* **Empirical Analysis**
*SOTA VLMs are evaluated, their weaknesses are extensively analyzed* (PFwd); *thorough, revealing significant performance gaps in models* (GnhX); *Valuable insight on cross-domain integration failures* (UKAL)
* **Clear Presentation**
*Well-written and easy to understand, with clear charts and diagrams* (YrMv), *Clear visualizations and well-structured presentation* (GnhX), *Provide insights into designing methods for real-world applications* (PFwd)

# Additional Experiments
* **Viewpoints (PFwd)** Comparison between first-person view and third-person view.
* **Hybrid planning methods (GnhX)** Three methods that integrate classical planning with LLMs: GPT-HTN, Neuro-Symbolic and Multi-Agent LLMs.
* **Training/Adaption (UKAL)** We add training-based methods and few-shot experiments.
* **Human Evaluation**  for reasoning quality, interaction naturalness, long-horizon planning and overall.
* **VLA models (YrMv)** We experiment with VLA models that output low-level actions.

*We sincerely thank all reviewers for their insightful feedback. All additional experiments will be reflected in the final version.*

Best,
Authors

---

### Decision · Program_Chairs · 2025-09-18

**Decision:**

Accept (spotlight)

**Comment:**

**Summary.** The paper presents Embodied Web Agents, a unified simulation platform that integrates 3D indoor (AI2-THOR), outdoor (Google Earth) and interactive web environments, and a ~1.5k-task benchmark spanning cooking, navigation, shopping, tourism, and geolocation. Baselines with leading LLM agents reveal substantial gaps to human performance, especially when tasks require cross-domain integration or coordinating physical actions with web-scale reasoning.

**Strengths.** Reviewers consistently praised the novelty of unifying embodied and web environments and the breadth/realism of the benchmark. The analyses clearly expose cross-domain integration as a failure mode for current agents. The authors added further analyses (e.g., viewpoints, hybrid planning, training/few-shot, VLA control, human evaluation) during discussion.

**Weaknesses.** Documentation is thin. I agree with reviewer GnhX that, even on GitHub, instructions are brief even when the authors link to claimed "very detailed instructions" and it is not obvious how to use the dataset or replicate the showcased agent control, which limits  utility. Simulation-only scope and reliance on custom sites are known trade-offs but acceptable for D&B.

**Discussion & rebuttal.** Authors provided clarifications, quantitative analyses (including an occlusion trade-off study), and the additional experiments listed above, and reiterated public availability (website, GitHub, HF). Reviewer GnhX raised their score after I inquired why it remained unchanged given that several concerns had been addressed. They nonetheless reiterated the documentation shortfall, which I share. The other reviewers maintained enthusiastic accepts but offered limited insight on usability; consequently, I reviewed the repository and documentation myself.

**Final recommendation justification** This is a substantial, timely contribution that opens a new evaluation space at the embodiment-web intersection, with strong reviewer support and added experiments. In the Datasets & Benchmarks track, the bar centers on artifacts that **enable or accelerate ML research** and whose code/data are **accessible, documented, and executable** (as required in the CFP). The current documentation outlines the commands to host the website, but is too brief to guide end-to-end setup and agent control, which **limits near-term utility** and thus the extent to which it accelerates community research. With substantially strengthened docs (end-to-end quickstart, runnable examples, reference agent scripts, usage recipes), this would be a strong Spotlight candidate.

===== FINAL UPDATE FROM DB Track PCs ====

The final decision for this paper has been taken by the program chairs after consultation with the SACs. All Senior Area Chairs have ranked papers according to the feedback from the AC during the review process. We decided to leave the original meta-review to reflect the opinion of the AC in light of the initial discussions with reviewers and SAC.